# Evidence of a persistent altered neural state in people with fibromyalgia syndrome during functional MRI studies and its relationship with pain and anxiety

**Patrick W. Stroman**[1,2,3]*, **Roland Staud**[4], **Caroline F. Pukall**[1,5]

**1** Centre for Neuroscience Studies, Queen's University, Kingston, Ontario, Canada, **2** Department of Biomedical and Molecular Sciences, Queen's University, Kingston, Ontario, Canada, **3** Department of Physics, Queen's University, Kingston, Ontario, Canada, **4** Division of Rheumatology, Department of Medicine, University of Florida, Gainesville, Florida, United States of America, **5** Department of Psychology, Queen's University, Kingston, Ontario, Canada

* stromanp@queensu.ca

**Data Availability Statement:** All relevant data for this study are publicly available from two figshare repositories: "Brain fMRI data comparing pain

## Abstract

Altered neural signaling in fibromyalgia syndrome (FM) was investigated with functional magnetic resonance imaging (fMRI). We employed a novel fMRI network analysis method, Structural and Physiological Modeling (SAPM), which provides more detailed information than previous methods. The study involved brain fMRI data from participants with FM (N = 22) and a control group (HC, N = 18), acquired during a noxious stimulation paradigm. The analyses were supported by fMRI data from the brainstem and spinal cord in FM and HC, brain fMRI data from participants with provoked vestibulodynia (PVD), and eye-tracking data from an fMRI study of FM. The results demonstrate differences in connectivity, and in blood oxygenation-level dependent (BOLD) responses, between FM and HC. In the FM group, BOLD signals underwent a large increase during the first 40 seconds of each fMRI run, prior to the application of any stimuli, compared to much smaller increases in HC. This indicates a heightened state of neural activity in FM that is sustained during fMRI runs, and dissipates between runs. The exaggerated initial rise was not observed in PVD. Autonomic functioning differed between groups. Pupil sizes were larger in FM than in HC, and the groups exhibited pupil dilation to the same levels during noxious stimulation. The initial BOLD increase varied in relation to state and trait anxiety scores. The results indicate that people with FM enter a heightened state of neural activity associated with anxiety and autonomic functioning, during every fMRI run, concurrent with increased pupil sizes, and heightened pain sensitivity. These findings may relate to the well-known hypervigilance and global hypersensitivity of FM participants.

## Introduction

Fibromyalgia (FM) is a prevalent chronic pain condition that affects 2–4% of the population [1]. Several theories about the underlying mechanisms have been proposed, including that it is

responses to noxious heat in healthy controls (HC) for comparison with fibromyalgia syndrome (FMS)" (https://doi.org/10.6084/m9.figshare.27105808.v1) and "Brain fMRI data comparing pain responses to noxious heat in fibromyalgia syndrome (FMS)" (https://doi.org/10.6084/m9.figshare.27103864.v1).

**Funding:** This work was supported by funding from the Canadian Institutes of Health Research [Project Grant 2023-2028] to PWS, RS, and CFP, and the Natural Sciences and Engineering Research Council of Canada [RGPIN-2020-06777] to PWS. The funders had no role in study design, data collection and analysis, decision to publish, or preparation of the manuscript.

**Competing interests:** The authors have declared that no competing interests exist.

mediated by central sensitization, motivational-affective aspects of pain, neurohormonal dysregulation, heightened stress responses, or autonomic dysregulation [2–8]. In addition, proposed mechanisms of altered endocrine function include dysregulation of norepinephrine [6], diminished serotonin transporter function [7], hyperactivity of the hypothalamic-pituitary-adrenal (HPA) axis, or a combination of hormonal systems [4]. A mathematical model has been proposed that describes an altered state of neural signaling in FM, involving the thalamus, somatosensory cortex, and HPA axis [9, 10]. The evidence to date consistently indicates that networks of brain and brainstem regions have altered function in FM.

Functional magnetic resonance imaging (fMRI) studies have provided new insights into altered neural signaling in FM in brain and brainstem regions [11–20]. Craggs et al. [18] showed evidence of central sensitization by means of connectivity analyses in brain regions. They also found that fMRI responses in the brain were similar with calibrated equivalent pain for participants with FM compared to healthy controls (HC). A later fMRI study in the brain, brainstem, and spinal cord supported the finding that BOLD response magnitudes were similar in brain regions with equivalent pain, but were different in some brainstem regions [16]. Differences in brainstem/spinal cord network connectivity were also identified between FM and HC groups with structural equation modeling (SEM) [13]. The regions included the locus coeruleus (LC), hypothalamus, periaqueductal gray region (PAG), and parabrachial nuclei (PBN), which are associated with autonomic regulation. Connectivity in brain regions identified in a condition without a pain stimulus (No-Pain condition) suggested that FM involves altered emotional responses to pain relief [21]. Connections identified in a condition with a noxious heat stimulus (Pain condition) indicate that FM may be associated with a dampened autonomic/fear response to upcoming noxious stimuli [21]. The data from this previous study are included in the present study, using different analysis methods. A concurrent study in the same participants investigated the brainstem and spinal cord and observed connectivity values that were correlated with pain ratings [13]. Several connections also had significant differences between the FM and control groups, including the locus coeruleus (LC), hypothalamus, periaqueductal gray region (PAG), and parabrachial nuclei (PBN). These regions are known to be associated with autonomic homeostatic regulation, including fight or flight responses. These findings are supported by a comparison of women with FM and with provoked vestibulodynia (PVD), which is a condition of idiopathic chronic vulvar pain [22, 23]. The results suggested that altered autonomic and pain regulation networks are specific to FM. Recently, Staud et al. [24] concluded that while brainstem activity is altered in FM and is likely related to hypersensitivity, central pain modulation showed no significant abnormalities in FM. These findings suggest that FM patients are hyperalgesic but modulate nociceptive input as effectively as HC.

The prior studies support the expectation that altered neural signaling in FM involves altered connectivity across regions in the brain, brainstem, and spinal cord. We have developed a novel fMRI analysis method, Structural and Physiological Modeling (SAPM), which provides more detailed information about neural signaling across networks than was previously possible [25, 26]. In the present study we apply SAPM to analyze fMRI data from previous FM studies spanning the brain, brainstem, and spinal cord, in the Pain and No-Pain conditions described above [13, 14, 20]. The objective is to obtain new insights into how neural signaling involved with nociception and pain is altered in women with FM.

## Methods

The data used for this analysis were obtained in prior studies in our lab as described above [13, 21]. All procedures were reviewed and approved by our institutional human research ethics board. The novel component of the present study is the application of our recently developed

analysis method (SAPM). The data were obtained from participants who had previously been diagnosed by a physician as having FM, and also participants without chronic pain as a control group. Data collection involved two visits, the first of which included an initial training session. This was followed by fMRI of the brainstem and spinal cord in one visit, and fMRI of the brain in another visit. The order of the fMRI sessions was alternated across participants to avoid order effects. The methods have been described in detail previously [13, 21] and are briefly repeated here.

## Participants

Participants in the prior studies were recruited from the local community and included a total of twenty-two women with previous diagnoses of FM and 18 healthy women [13, 21]. All participants provided informed written consent prior to participating. For the present analysis, the data were accessed between June 2023 and March 2024, and were in anonymized form and the participants could not be identified. The study included two MRI sessions, approximately one week apart, with fMRI data being acquired from the brain in one session, and brainstem and spinal cord in another session. The order of the studies was randomized across participants. The participants with FM were identified based on having a prior diagnosis by a physician, and based on pressure-point testing with an algometer, and questionnaires as described below. Participants with FM were not asked to withhold their medications, but were included in the study only if their medications had been stable for at least 3 months. Of the total number of participants recruited, complete brain fMRI datasets were obtained from 20 FM (age range = 24–64, $M_{age}$ = 48.8 ± 12.7; mean ± SD) and 17 HC (age range = 21–59, $M_{age}$ = 37.9 ± 10.4; mean ± SD) participants. However, not all participants completed both sessions and complete fMRI data sets from the spinal cord and brainstem were obtained from 15 women with FM (mean age 46 ±13 years) and 15 healthy women (mean age 39 ±10 years). Data collection was subsequently interrupted by the COVID-19 pandemic. The participants are characterized in detail in the original paper [21]. Relevant to the analyses described below is that of the 20 participants with complete brain fMRI datasets, ten were taking serotonin and norepinephrine reuptake inhibitors (SNRIs), whereas the remaining ten were not taking medications that directly influence norepinephrine levels. Two of the 20 FM participants were taking opioid antagonists, one was taking opioid agonists, two were taking anxiolytic medications, and 17 of the 20 were on some form of antidepressant.

## Questionnaires

Participant characteristics were identified by means of a series of questionnaires. These included the State-Trait Anxiety Inventory (STAI) [27], the Beck Depression Inventory-II (BDI-II) [28], and the Composite Autonomic Symptom Score 31 (COMPASS-31) [29]. Participants also completed the Social Desirability Scale (SDS) [30] and the Pain Catastrophizing Scale (PCS) [31] and the Short-Form McGill Pain Questionnaire-2 (SF-MPQ-2) [32]. Participants with FM also completed the Revised Fibromyalgia Impact Questionnaire (FIQR), and control participants completed the corresponding SIQR questionnaire [33]. Finally, because FM inclusion was based on a prior diagnosis by a physician, we included the 2016 Fibromyalgia Survey Questionnaire (FSQ) to determine which participants (FM or HC) currently met the most recent classification criteria for FM [34].

## Participant training

During the first study visits, participants were first familiarized with the study procedures and the thermal stimulus used to evoke pain. For participants in the FM group this session

included a tender-point examination by means of an algometer, as detailed previously [21]. Participants were then familiarized with a 0–100 numerical pain rating scale (NPS) which they subsequently used to rate their pain. They were then familiarized with the custom-made MRI-compatible thermal stimulator (RTS-2). This device was used to administer noxious heat stimuli to the hand, by means of a series of brief repeated contacts of a heated thermode, under precise timing and temperature control, as described previously [13, 21].

A series of tests were carried out to familiarize the participants with the heat stimuli and the study paradigm, and to calibrate the temperature to evoke moderate pain (approximately 50/100) in each participant. For each test the thermode temperature was set, and then the participant experienced the thermode contacting the skin on their hand repeatedly for 1.5 seconds, with onsets every 3 seconds. Initial tests involved 3 contacts, and then a series of tests with 10 contacts were carried out at different temperatures (46˚C, 50˚C, 44˚C, and 48˚C). Participants were asked to verbally rate their pain from each contact of the thermode. These tests demonstrated the degree of temporal summation of second pain experienced during each set of contacts, and enabled us to calibrate the temperature for each participant, while also enabling the participant to become familiarized with the procedures. Following these tests, participants were positioned in a mock-up of the MRI system and experienced one practice fMRI run of the complete stimulation paradigm. For this run they were asked to mentally rate their pain to each contact, and to remember their ratings for the first and last contact. At the end of the practice run they were asked to verbally report their ratings. The calibrated temperature was used in subsequent MRI sessions for each participant.

## Stimulation paradigm

Each fMRI acquisition employed our "threat/safety" paradigm as described previously [13, 21, 35]. This involved two types of runs, one with a period of noxious stimulation (Pain) and the other with no stimulation (No-Pain). Participants viewed a rear-projection display while positioned inside the MRI system and were given cues as to when a run would start, information during the run about the stimulus to expect, and other than when information was being provided the display showed the pain rating scale. During the first minute of each run, participants were unaware of which type of paradigm they would experience. At the 1-minute mark they were informed whether they would experience a noxious stimulus, or not. If a noxious stimulus was to be applied, it began at the 2-minute mark and consisted of 10 heat contacts of 1.5 seconds duration and onsets every 3 seconds, as practiced in the training session. Repeated heat contacts were used as the stimulus in order to cause temporal summation of second pain (TSSP) and demonstrate the effects of central sensitization that are believed to occur in FM [16, 36]. The thermode temperature was set to the temperature determined to evoke moderate pain during the training session for the participant. Following the 30 seconds of noxious stimulation, data collection continued for another 2 minutes, for a total run duration of 4.5 minutes. If no stimulus was to be applied, the acquisition continued for a total of 4.5 minutes with no stimulus being applied. Participants were able to anticipate the timing and experience of each paradigm as a result of the initial training session. Participants were reminded at the start of the fMRI session to mentally rate their pain to each contact as practiced during the training session. At the end of each run participants were asked to verbally report their pain rating for the first and last of the heat contacts. As described previously [21], the calibrated temperature was significantly different ($p = 2.55 \times 10^{-5}$) between the two groups and averaged 50.89 ± 1.04˚C for the HC group and 47.51 ± 2.72˚C for the FM group. The average pain ratings to the last contact in each set of 10 contacts were 38.93 ± 12.07 and 44.22 ± 12.04 on a

0–100 scale, for the HC and FM groups respectively, and were not signficantly different. The temperatures and pain ratings demonstrate the higher heat pain sensitivity in the FM group.

## Functional MRI data acquisition

Image data were acquired using a 3 tesla whole-body MRI system. Approximately midway through data collection, the MRI system was upgraded from a Siemens Trio to a Siemens Prisma (Siemens, Erlangen, Germany). Test scans involving our experimental paradigm were performed on two participants before and after the upgrade, and the data quality were compared. No significant differences in BOLD activity between the datasets were found. Of the 20 FM participants included in this study, 15 were studied before the MRI upgrade, and 5 were studied after the upgrade. Of the 17 HC participants, 12 were studied before the upgrade, and 5 after. Participants were positioned supine and were supported by foam padding as needed to ensure comfort and minimize bulk body movement. Structural images were acquired at the beginning of the scanning session using a sagittal, $T_1$-weighted MPRAGE sequence (TR = 1760 ms, TE = 2.2 ms, Inversion Time = 900 ms, Flip Angle = 9˚, Resolution = 1 x 1 x 1 mm$^3$). To relate brain and brainstem/spinal cord fMRI data, the imaging window spanned from the top of the C1 vertebra to the top of the cortex. FMRI data were acquired in 66 contiguous axial slices using a GE-EPI sequence (TR = 2 s, TE = 30 ms, Flip Angle = 84˚, simultaneous multi-slice (i.e., multiband) factor of 3, 7/8 partial Fourier sampling of k-space, FOV = 180 mm x 180 mm, matrix = 90 x 90, resolution = 2 x 2 x 2 mm$^3$). A 32-channel head coil was used for signal reception, and a body coil was used for transmission of RF pulses. A total of 135 volumes were acquired for each imaging run of 4.5 minutes, and each participant experienced 4 to 5 Pain and 4 to 5 No-Pain runs.

FMRI data in the brainstem and spinal cord were acquired using our established methods with a half-Fourier single-shot fast spin-echo (HASTE) sequence with BOLD contrast [37]. Images spanned the entire brainstem and cervical spinal cord (first thoracic vertebra to above the thalamus). This method has been shown to provide optimal image quality and BOLD sensitivity in the brainstem and spinal cord. The 3D volume was imaged in 9 contiguous sagittal slices, 2 mm wide, with a 28 x 21 cm field-of-view and a 1.5 x 1.5 mm in-plane resolution. Imaging parameters included an echo time (TE) of 76 ms and a repetition time (TR) of 6.75s/ volume for optimal $T_2$-weighted BOLD sensitivity. Each imaging run consisted of 40 volumes (total of 4.5 minutes per run). In total, 10 runs were acquired for each participant, 5 Pain and 5 No Pain, and each condition consisted of 200 volumes per individual.

## Data analysis

All fMRI data analyses were carried out using the Pantheon software package in python [26] (https://github.com/stromanp/pantheon-fMRI). Data were first converted from DICOM to NIfTI format and pre-processed. Pre-processing steps included motion correction (co-registration), slice-timing correction, spatial normalization, and removal of physiological noise. Initial time points with variable $T_1$-weighting were removed and replaced with values from the first subsequent time point. Points were replaced in order to preserve information about the time lapsed since the beginning of the fMRI run. For brain fMRI data the first 3 points were replaced, and for spinal cord and brainstem data the first 2 points were replaced.

Spatial normalization was guided by reference images corresponding to the MNI152 template for data from the brain, and the extension of this template into the brainstem and spinal cord as described previously [37]. The reference images were created by combining the MNI152 template from the Statistical Parametric Mapping (SPM12) software package [38] and the PAM50 template, as described by De Leener et al. [39]. Anatomical region-of-interest

maps were defined from multiple sources, including anatomical descriptions, freely shared anatomical maps, and the CONN15e software package [40–50]. These sources were combined to create a single anatomical map. The normalization procedure employed different methods for the brain, and for brainstem and cord regions. For brain regions, the normalization process uses the python package "dipy" (https://dipy.org/documentation/1.5.0/documentation/) which is based on the ANTs (Advanced Normalization Tools) software [51, 52]. Normalization of brainstem and spinal cord regions has been described previously [26, 53] and involves mapping sections of the template to the image data for brainstem regions with distinct anatomical features. These region positions then guide the identification of successive sections of the spinal cord based on cross-correlation, progressing away from the brainstem, in a manner that maintains the distance along the cord. The normalization procedure is based on the premise that the cord anatomy is likely much more consistent in size across people, than is the spine anatomy [44]. The normalization was then fine-tuned using the Medical Image Registration (MIRT) toolbox, translated into python [54].

Following pre-processing, voxels within selected anatomical regions within a predefined network were identified and BOLD responses were extracted for each fMRI run. A k-means clustering method was used to identify voxels within 5 sub-regions of approximately equal volume within each region. The BOLD time-courses of the sub-regions were used to identify features of the responses in relation to periods within the stimulation paradigm, as well as for connectivity analysis by means of Structural and Physiological Modeling (SAPM).

SAPM has been described in detail and validated previously [25, 26]. It is essentially a connectivity analysis method and is an extension of structural equation modeling (SEM). However, SAPM combines information about BOLD responses, physiology, and anatomical information, to model the input and output signaling from every region in a pre-defined network in a way that explains the observed BOLD responses. The network model used for the current analysis was based on known neuroanatomy involved with nociceptive processing and pain, and based on our prior studies employing SEM analyses [21, 42, 55]. The model (Fig 1) includes the anterior cingulate (AC), posterior cingulate (PC), insular cortex (IC), heschl's gyrus (HG), the frontal orbital cortex (F. Orb.), hippocampus (Hippo.), hypothalamus (Hypo.), amygdala (Amyg.), thalamus (Thal.), nucleus accumbens (N. Accum.), ventral tegmental area (VTA), periaqueductal gray region (PAG), parabrachial nuclei (PBN), and locus coeruleus (LC). It also includes latent inputs from outside of the network to the frontal orbital cortex, insular cortex, and locus coeruleus (labeled Lat0, Lat1, Lat2, respectively). The corresponding network model used for analyzing data from the brainstem and spinal cord has been described in detail previously [26, 56]. This brainstem/cord network includes the thalamus, hypothalamus, PAG, LC, PBN, nucleus tractus solitarius (NTS), regions within the rostral ventromedial medulla (RVM) including the nucleus gigantocellularis (NGc) and nucleus raphe magnus (NRM), the dorsal reticular nucleus of the medulla (DRt), and the spinal cord dorsal horn in 6th cervical segment on the right side of the body (C6RD).

With SAPM the input signaling to each region, $S_{input}$, is modeled as the sum of the outputs from other regions in the network:

$$S_{input} = M_{input} \, S_{output}$$

In this equation $M_{input}$ is a matrix of "D" values which are the weightings of each incoming signal which sum to the total input signaling to each region, and $S_{output}$ is the output signaling from each region. All D values are positive and allow for the output signaling from a single region to contribute different amounts of input to other regions. The output signaling from

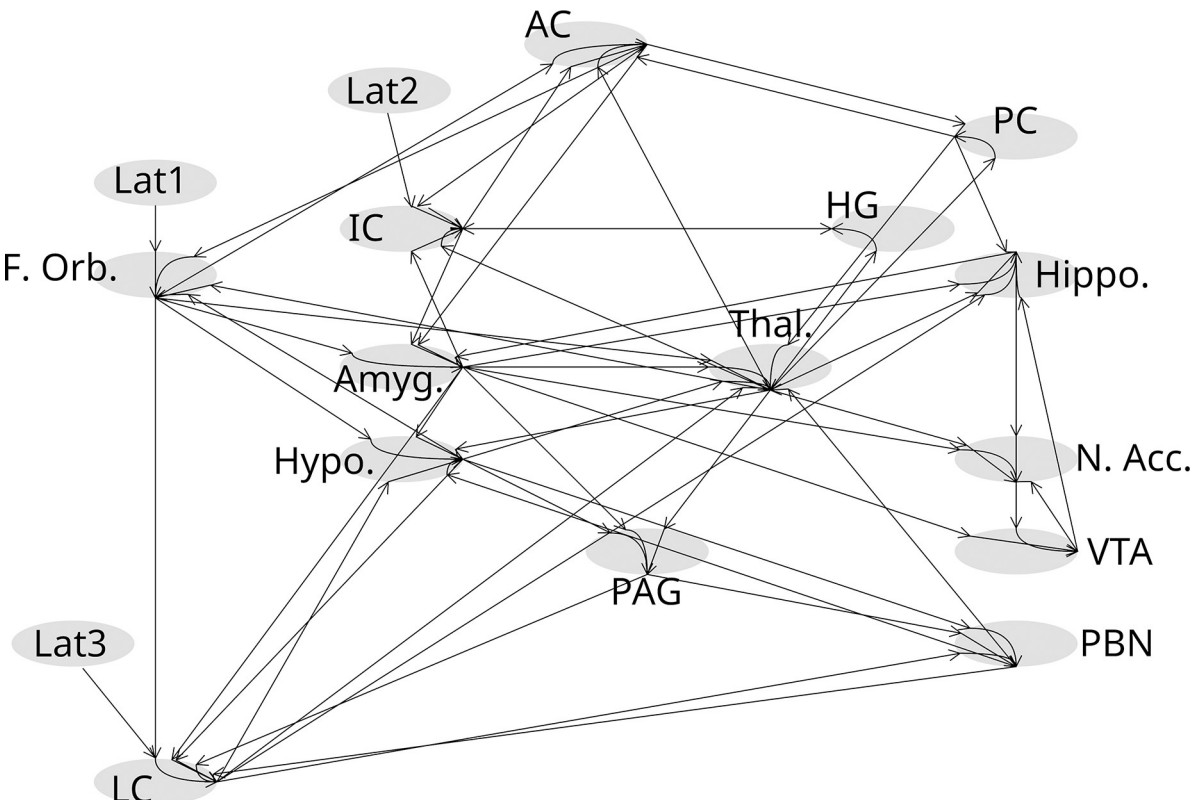

**Fig 1. Graphical representation of the network model used for SAPM analysis of data from brain and brainstem regions.** Regions are represented by filled ovals, and lines indicate the direction of possible signaling between regions. Abbreviations are defined in the text.

each region is modeled similarly as:

$$S_{output} = M_{output} \ S_{output}$$

Here, $M_{output}$ is a matrix of weighting factors that are related to how each input signal influences the output signaling from the region. A positive weighting factor corresponds with excitatory input (more input produces more output). A negative value corresponds with inhibitory input (more input produces less output). The weighting factors in $M_{output}$ are termed "DB" values because they are the product of the D values and B values. The B values reflect how the incoming signal is converted to contribute to the output signal within each region. The B and DB values demonstrate the "apparent transmission effect" of each region, for each incoming signal [26].

SAPM was applied to determine the network values for a selected set of sub-regions (one sub-region per region). The results explain the coordinated signaling across the network and the proportion of variance in each sub-region that can be explained by the network model. The statistical significance of group-average connectivity values was determined by comparing values with the results of "null" simulations using 10,000 sets of simulated data composed of normally-distributed random values. The results can also be used to identify which combinations of sub-regions best fit the network model, and thus provide anatomical information. A gradient-descent search method was used, starting from a random selection, and iteratively vary the sub-regions until the combination with the least fit error was found. This combination of sub-regions therefore has BOLD time-series responses that fit the network model better

than other combinations of sub-regions. This same set of sub-regions was then used for all subsequent analyses for consistency. In order to compare results from the brain, and from brainstem/cord, the same sub-regions were used for SAPM analyses for the regions that are common to both network models.

The network parameters were determined with SAPM using data from each participant, and thus demonstrate variations with individual pain responses, and in relation to participant characteristics. The values were also analyzed in relation to the study group (FM vs HC). Finally, the BOLD time-course responses in each sub-region which best fits the network model were analyzed to identify group-average features, as well as where signal variations are correlated with pain ratings or questionnaire scores. The time-series responses also compared with the timing of the stimulation paradigm, and were compared between FM and HC study groups.

### Additional data sets for validating results

Some of the findings were verified by comparing results from additional data sets obtained from human research participants. For these studies the recruitment procedures were as described above, and as reported previously [22, 57]. One set of data was from a study of female participants with PVD [23]. The data were accessed between February 2024 and March 2024 and were in anonymized form. The study group consisted of 14 participants with PVD (age 31 ± 10 years) and 14 healthy female participants without any pain conditions (age 31 ± 10 years). The study procedures were identical to those described for the FM study group.

Eye-tracking data were obtained from a currently on-going fMRI study in our lab that involves the same stimulation paradigm and study procedures as the prior FM and PVD studies. All study procedures were approved by our institutional research ethics board, and data used for this study were collected between May 2023 and March 2024, and were in anonymized form. Eye-tracking data were recorded during fMRI acquisitions using an SR Eyelink1000 eye-tracking system (SR Research Ltd, Ottawa, Canada), with data recorded at 500 Hz. This data set consisted of 125 trials in 26 female participants with FM (age 47 ± 11) and 45 trials in 10 female participants without chronic pain (age 43 ± 14). At the start and end of each trial viewed a completely white screen followed by a completely black screen to provide pupil size references.

## Results

A number of significant connections were identified based on group average connectivity (DB) values compared to values obtained with "null" data simulations (Table 1). Significance is inferred at a family-wise error corrected $p < 0.05$, which corresponds with an uncorrected $p < 0.00156$, and $|T| > 3.53$. These values are seen to be predominantly positive (indicating excitatory signaling) with a few notable exceptions. The values are also seen to be relatively consistent across the four group/condition combinations that were tested.

The SAPM results identify specific anatomical sub-regions which have BOLD responses that best fit the model network. These sub-regions were used for the SAPM analysis and the same sub-regions were used for both data sets (brain and brainstem/cord) for consistency. A comparison of the BOLD responses between the two data sets is shown in supplementary information (S1 Appendix), for the locus coeruleus. This comparison demonstrates that the BOLD responses provide similar information in the two data sets and that the SAPM results can be combined. The network results are therefore plotted together to span the brain, brainstem and spinal cord, in Fig 2.

Analyses of covariance (two-way ANCOVAs) were used to investigate how connectivity strengths (specifically DB values) varied in relation to the study groups (FM vs HC) and

**Table 1. Summary of group average connectivity (DB) values for the FM and HC groups, in Pain and No-Pain conditions.** Values are listed as the mean ± standard error of the mean. Significant values are shown in bold-face font, and are inferred at a family-wise error corrected p < 0.05, which corresponds with an uncorrected p < 0.00096, and |T| > 3.55.

**Brain Data**

| Connection | FM Pain DB | T | FM No-Pain DB | T | HC Pain DB | T | HC No-Pain DB | T | Notes |
|---|---|---|---|---|---|---|---|---|---|
| AC→PC | **0.402 ± 0.069** | **6.08** | **0.580 ± 0.077** | **7.77** | **0.679 ± 0.066** | **10.58** | **0.471 ± 0.070** | **6.96** | lower in FM during Pain |
| IC→AC | **0.679 ± 0.155** | **4.44** | **0.687 ± 0.168** | **4.13** | **0.732 ± 0.185** | **3.99** | **0.799 ± 0.124** | **6.51** | lower in FM |
| AC→IC | **0.275 ± 0.072** | **4.24** | **0.393 ± 0.059** | **7.13** | **0.459 ± 0.073** | **6.66** | **0.392 ± 0.052** | **8.04** | lower in FM during Pain |
| Thal→AC | **0.276 ± 0.073** | **3.96** | 0.223 ± 0.118 | 2.00 | 0.175 ± 0.098 | 1.91 | 0.131 ± 0.079 | 1.82 | higher in FM |
| Amyg→Thal | **0.311 ± 0.085** | **3.90** | **0.457 ± 0.095** | **5.06** | **0.547 ± 0.113** | **5.01** | **0.381 ± 0.112** | **3.58** | lower in FM during Pain |
| Thal→PC | **0.326 ± 0.094** | **3.86** | 0.184 ± 0.097 | 2.28 | 0.144 ± 0.113 | 1.60 | 0.249 ± 0.088 | 3.24 | higher in FM during Pain |
| PC→AC | **0.133 ± 0.037** | **3.81** | **0.295 ± 0.051** | **5.97** | **0.285 ± 0.066** | **4.43** | **0.226 ± 0.036** | **6.55** | lower in FM during Pain |
| AC→FOrb | **-0.253 ± 0.067** | **-3.77** | **-0.229 ± 0.058** | **-3.92** | -0.239 ± 0.124 | -1.92 | **-0.414 ± 0.091** | **-4.54** | FM has similar values to HC Pain |
| PBN→Thal | **0.094 ± 0.032** | **3.55** | -0.009 ± 0.038 | 0.28 | 0.113 ± 0.060 | 2.20 | -0.020 ± 0.034 | -0.00 | higher in Pain than No-Pain |
| Hypo→PAG | 0.036 ± 0.108 | 0.55 | -0.033 ± 0.087 | -0.12 | **0.252 ± 0.054** | **5.06** | 0.209 ± 0.087 | 2.67 | lower in FM |
| Thal→Hypo | 0.245 ± 0.073 | 3.50 | 0.250 ± 0.077 | 3.36 | 0.257 ± 0.094 | 2.86 | **0.406 ± 0.062** | **6.73** | FM has similar values to HC Pain |
| Amyg→Hipp | 0.098 ± 0.095 | 1.11 | -0.053 ± 0.113 | -0.41 | 0.291 ± 0.108 | 2.74 | **0.423 ± 0.111** | **3.86** | lower in FM |

**Connections of interest not listed above that occur in both brain data and brainstem/cord data (note that thalamus clusters are not the same in brain and brainstem/cord data)**

| Connection | FM Pain DB | T | FM No-Pain DB | T | HC Pain DB | T | HC No-Pain DB | T | Notes |
|---|---|---|---|---|---|---|---|---|---|
| LC→PBN | -0.050 ± 0.122 | -0.83 | 0.150 ± 0.104 | 0.95 | -0.149 ± 0.151 | -1.33 | -0.087 ± 0.127 | -1.08 | lower in Pain than No-Pain |
| Hypo→Thal | 0.340 ± 0.100 | 3.47 | 0.190 ± 0.105 | 1.88 | 0.318 ± 0.132 | 2.47 | 0.171 ± 0.057 | 3.14 | higher in Pain than No-Pain |
| LC→Thal | 0.347 ± 0.104 | 3.39 | 0.110 ± 0.132 | 0.88 | 0.074 ± 0.108 | 0.74 | -0.037 ± 0.099 | -0.31 | higher in FM |

**Brainstem and Spinal Cord Data**

| Connection | FM Pain DB | T | FM No-Pain DB | T | HC Pain DB | T | HC No-Pain DB | T | |
|---|---|---|---|---|---|---|---|---|---|
| LC→Thal | **0.478 ± 0.113** | **4.50** | **0.581 ± 0.114** | **5.37** | **0.566 ± 0.116** | **5.17** | **0.736 ± 0.092** | **8.37** | FM has similar values to HC Pain |
| LC→PBN | 0.612 ± 0.189 | 2.63 | **0.775 ± 0.124** | **5.32** | 0.410 ± 0.168 | 1.76 | **0.721 ± 0.125** | **4.86** | lower in Pain than No-Pain |
| NRM→C6RD | -0.266 ± 0.104 | -2.68 | **-0.285 ± 0.081** | **-3.68** | -0.034 ± 0.062 | -0.76 | -0.050 ± 0.044 | -1.44 | higher mag. in FM than HC |
| Hypo→NTS | 0.054 ± 0.128 | 0.32 | **0.220 ± 0.058** | **3.57** | -0.002 ± 0.081 | -0.19 | -0.110 ± 0.080 | -1.53 | No clear pattern |
| PBN→Thal | 0.275 ± 0.135 | 2.19 | 0.241 ± 0.090 | 2.91 | **0.395 ± 0.093** | **4.47** | 0.309 ± 0.126 | 2.63 | lower in FM |
| Hypo→Thal | 0.197 ± 0.126 | 1.39 | 0.125 ± 0.091 | 1.12 | 0.360 ± 0.121 | 2.80 | **0.610 ± 0.128** | **4.59** | lower in FM |

questionnaire scores from each participant. All statistical analyses were done in python using custom-written software using freely shared python modules. Correlations were computed using the "corrcoef" function in the "numpy" (version 1.23.5) module, and ANCOVAs were computed using the "anova_lm" function in the "statsmodels" (version 0.14.0) module. Connectivity strengths were determined to vary between groups and in relation to questionnaire scores (Table 2). Specifically, the AC-IC connection was observed to differ between FM and HC groups when variations in COMPASS-31 scores were accounted for. Also, the AC-PC connection was observed to differ between FM and HC groups when variations in STAI-Y-1 scores were accounted for. The connections with the strongest significance were the IC-HG connection which varies in relation to COMPASS-31 scores, and the AC-Forb connection which varies with PCS Magnification scores in a way that differs between FM and HC groups (i.e. interaction effect).

The connectivity (DB) values demonstrate the relationship between input signaling to a region and how it contributes to the total output signaling from that region. With the SAPM method, variations in DB values across groups or conditions are expected to demonstrate differences in neural excitability of a region [25]. Variations in BOLD responses, reflecting the net total input signaling to each region, are a separate component of how neural signaling can

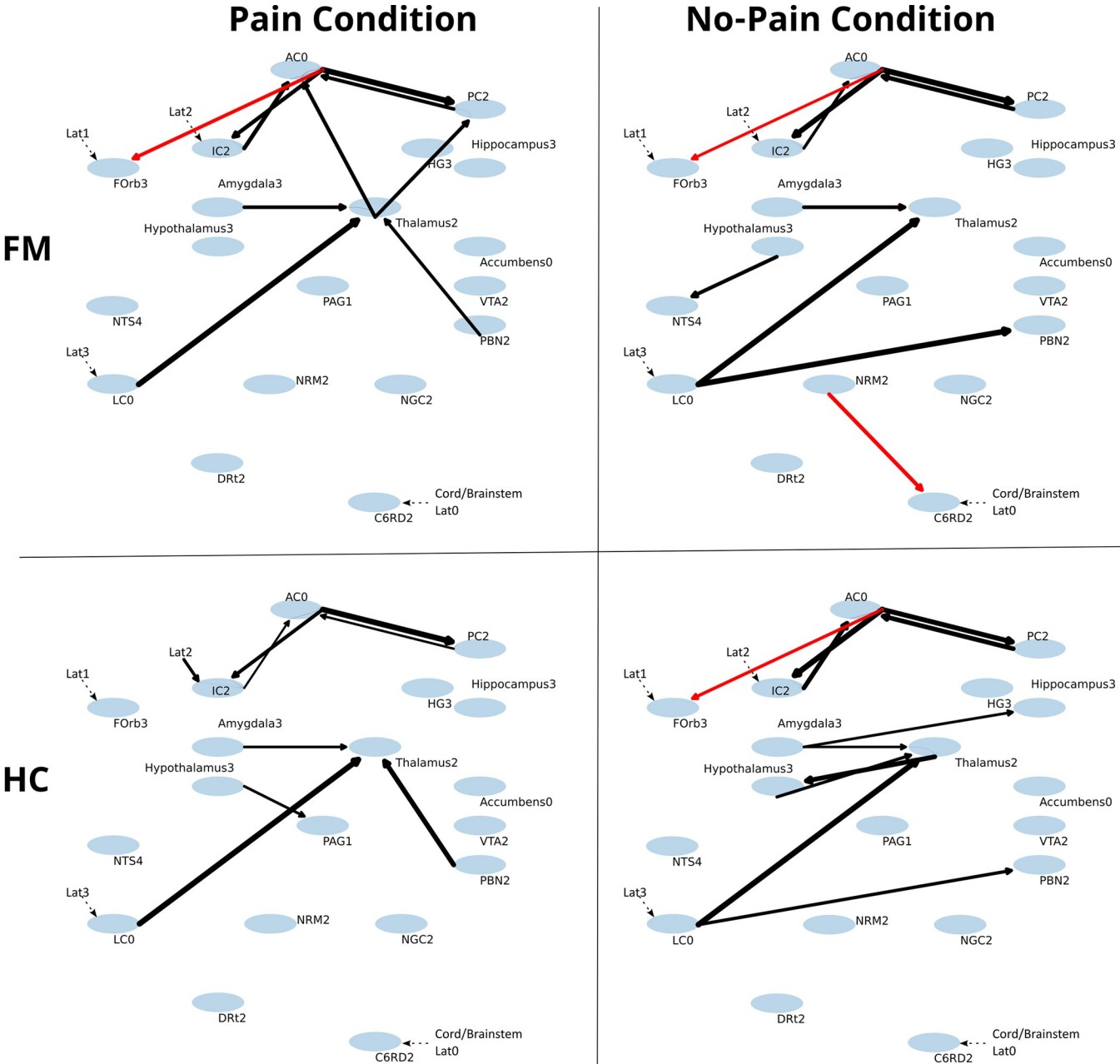

**Fig 2. Plots of connected regions corresponding to Table 1, obtained with SAPM analysis.** The upper row shows data obtained in participants with fibromyalgia (FM) and the lower row shows data from healthy participants (HC). Each plot demonstrates regions, indicated with gray oval, and significant connections indicated with lines, and arrows indicating the direction of influence. Black lines indicate positive connectivity values (excitatory effect) and red lines indicate negative connectivity values (inhibitory effects). Only significant connections are shown, based on T-test comparisons of group values with null results (p < 0.05 family-wise error corrected, p < 0.00156 uncorrected).

vary between groups and conditions. The observed BOLD responses were averaged across runs for all participants and are plotted in Fig 3 to demonstrate differences in features between Pain and No-Pain conditions, and between FM and HC groups across different regions.

The BOLD time-course responses in Fig 3 demonstrate signal variations at the time that participants were informed of whether or not they would feel a noxious stimulus, and also

**Table 2. ANCOVA results comparing participant characteristics and connectivity values (DB) between brain regions, across participants in FM and HC groups during the Pain condition ($p < 0.01$).** Family-wise error correction for the number of connections in the network requires $p < 0.00096$ (indicated in bold-face font).

| Covariate | Effect of: | Connection | FM (avg ± sem) | HC (avg ± sem) | p |
|---|---|---|---|---|---|
| **Pain Rating** | Covariate | AC-Amyg | 0.071 ± 0.156 | 0.146 ± 0.137 | $2.63 \times 10^{-3}$ |
| | Interaction | Thal-Hipp | 0.165 ± 0.115 | 0.296 ± 0.122 | $7.70 \times 10^{-3}$ |
| **COMPASS** | Group | AC-IC | 0.029 ± 0.029 | -0.042 ± 0.022 | $2.19 \times 10^{-3}$ |
| | Covariate | IC-HG | 0.340 ± 0.100 | 0.318 ± 0.132 | $1.57 \times 10^{-3}$ |
| | Interaction | Hypo-PAG | 0.245 ± 0.073 | 0.257 ± 0.094 | $6.32 \times 10^{-3}$ |
| **STAI-Y-1** | Group | AC-PC | 0.402 ± 0.069 | 0.679 ± 0.066 | $7.30 \times 10^{-3}$ |
| **PCS Rumination** | Interaction | Thal-PC | 0.326 ± 0.094 | 0.144 ± 0.113 | $3.98 \times 10^{-3}$ |
| **PCS Magnification** | Interaction | **AC-FOrb** | **-0.253 ± 0.067** | **-0.239 ± 0.124** | **$9.45 \times 10^{-4}$** |

during the stimulation period. Notable increases in BOLD signal occur after the onset of stimulation, and after the offset of stimulation. The BOLD responses to the Pain and No-Pain stimulation paradigms are shown to be similar in the first minute, prior to participants being informed of the stimulus, or not. During this period there is a notable rise in signal to the approximate level that is maintained for the remainder of the fMRI acquisition.

Details of the observed BOLD time-course responses were compared between participants in the FM and HC groups, and are plotted in Fig 4 for selected regions. The HC group responses are plotted with the average value equal to zero. However, given that the FM group responses have large variations during the first minute, they are plotted with the same starting values as for the HC group. This is to illustrate how the time-course responses compare if we are to assume that the two groups start from the same baseline state. However, the alternative possibility is considered in the Discussion section.

The initial rise in magnitude was investigated by comparing the BOLD responses across participants, in relation to a number of factors including the run number (i.e. does the response vary across repeated fMRI runs), pain ratings, and questionnaire scores. For brevity, only results which demonstrated apparent relationships are shown, but details of all ANCOVA comparisons are listed in supplementary information (S2 Appendix). The initial rise in magnitude varied considerably across participants, ranging from roughly 0.0 to -1.0 (express as % signal change). However, this magnitude was not observed to depend on the run number, and there are no clear relationships between the magnitude of the initial rise, any of the questionnaire scores, or pain ratings or pain sensitivity. In order to investigate the effects of noradrenaline, participants were compared between those taking SNRIs to those who were not. No difference in the original rise was detected with this comparison. Comparisons of BOLD time-course responses stratified into high, medium, and low pain sensitivity are shown in supplementary information (S3 Appendix), and demonstrate the consistency of the initial rise in the FM group with no apparent consistent dependence on pain sensitivity.

Relationships between the size of the initial rise and questionnaire scores were further investigated by means of analyses of covariance (ANCOVAs). The results are listed in Table 3 for results reaching significance at $p < 0.01$. For brevity, only the comparisons that showed potential relationships are listed, but details of all ANCOVA comparisons are listed in supplementary information (S2 Appendix). The connectivity values and scores are plotted for selected results in Fig 5 to illustrate the relationships between values. Accounting for variations in STAI-Y-1 scores, the most significant effect is that of the group (FM vs HC) in the FOrb, IC, hypothalamus, and PC. This means that the differences are between the groups and are not necessarily related to the STAI-Y-1 score. The amygdala, nucleus accumbens, and IC also have

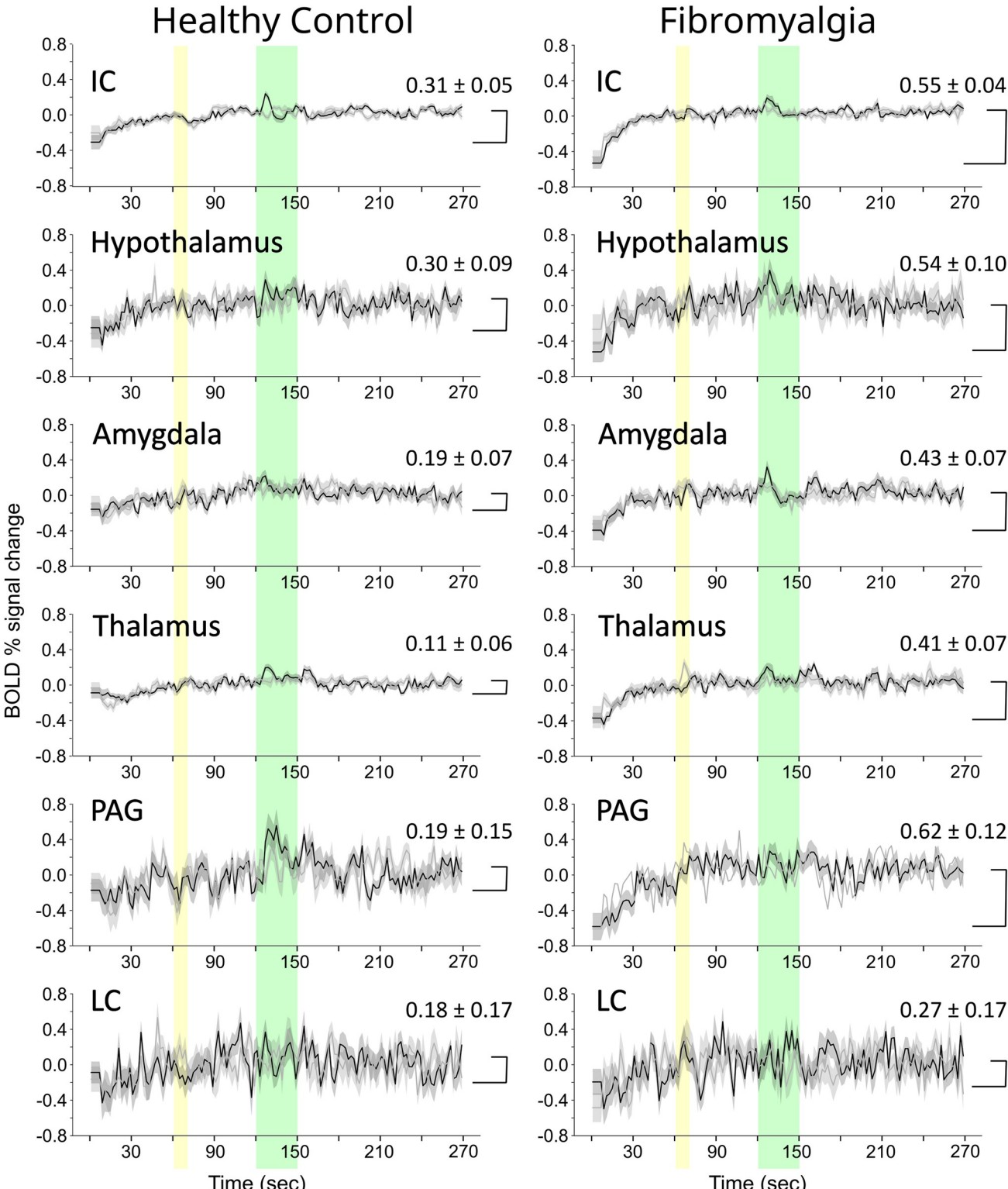

**Fig 3. Comparisons of BOLD time-course responses in selected regions from brain fMRI data.** Results from healthy participants are shown in the left column and results from participants with fibromyalgia and in the right column. Black lines indicate BOLD responses for runs with noxious stimulation, and gray lines show responses to runs without stimulation. The period when participants were informed of the stimulus type is indicated with a yellow band, and the stimulation period is indicated with a green band.

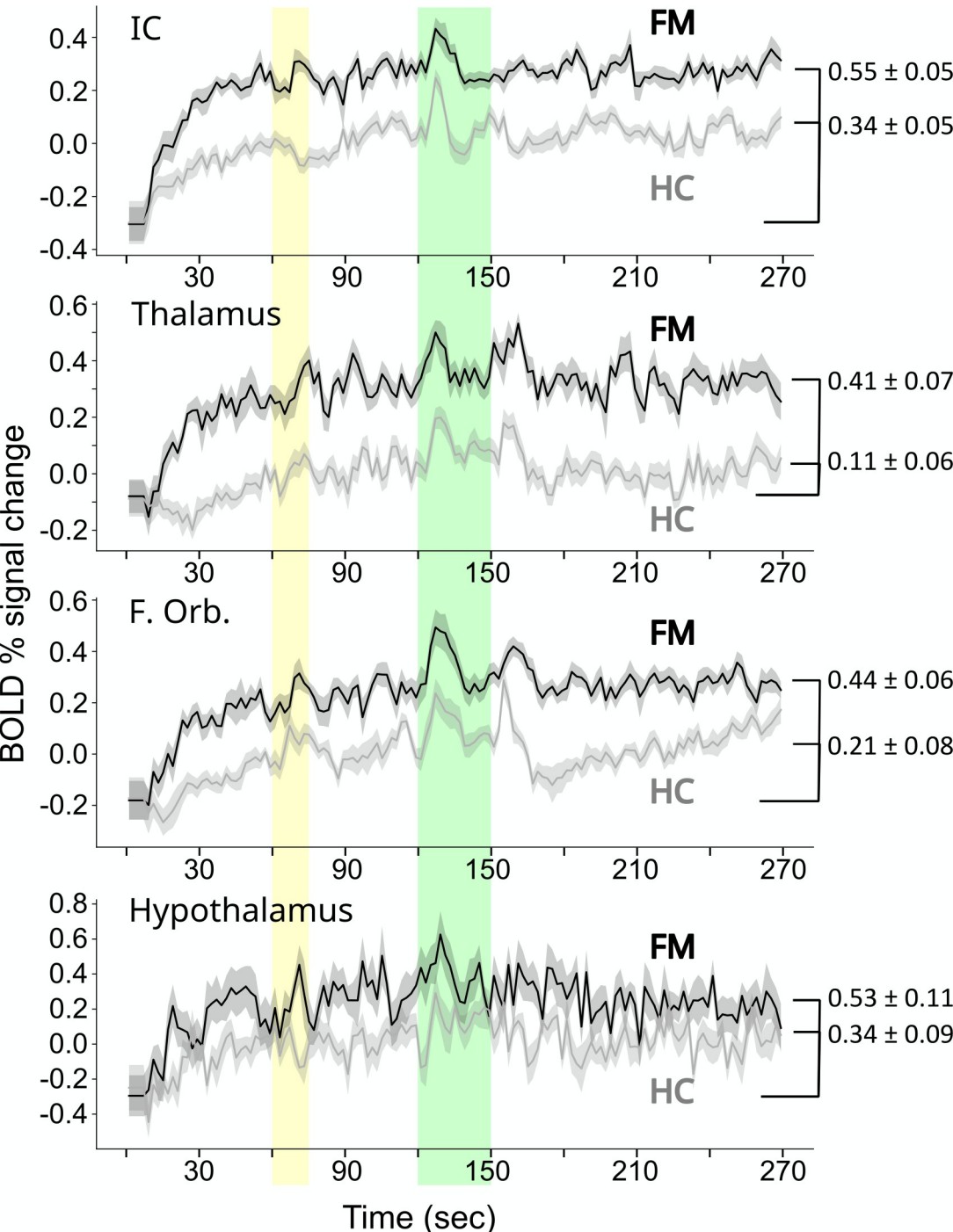

**Fig 4. Overlaid plots of BOLD time-course responses in selected regions, for FM (black lines) and HC (gray lines) groups, during the pain condition.** The plots are shifted in the vertical direction to align the starting values for the two groups. The time period corresponding to when participants were informed of the stimulus type (Pain or No-Pain) is indicated in yellow, and the period of noxious heat stimulation in indicated in green.

significant (p < 0.01) effects of the STAI-Y-1 score, but not significant interaction effects meaning that there are relationships between the initial rise and the score, but the relationship is not different between FM and HC participants. Accounting for variations in STAI-Y-2, the

**Table 3. ANCOVA results corresponding to Fig 5.** Values are shown with p < 0.01. Correcting for multiple comparisons based on the number of connections in the network requires $p < 9.6 \times 10^{-4}$.

| Region | Covariate | FM group ($R^2$ value) | HC group ($R^2$ value) | Main Effect of Group | Main Effect of Covariate | Interaction Effect |
|---|---|---|---|---|---|---|
| F. Orb. | STAI-Y-1 | 0.435 | 0.199 | **$8.05 \times 10^{-4}$** | $9.87 \times 10^{-2}$ | 0.700 |
| IC | STAI-Y-1 | 0.581 | 0.307 | **$2.55 \times 10^{-3}$** | **$5.82 \times 10^{-3}$** | 0.416 |
| N. Acc. | STAI-Y-1 | 0.446 | 0.191 | 0.366 | **$2.73 \times 10^{-3}$** | 0.683 |
| Hypoth. | STAI-Y-1 | 0.395 | 0.272 | **$3.30 \times 10^{-3}$** | $7.59 \times 10^{-2}$ | 0.634 |
| Amygdala | STAI-Y-1 | 0.257 | 0.292 | 0.135 | **$5.62 \times 10^{-3}$** | 0.805 |
| PC | STAI-Y-1 | 0.427 | 0.343 | **$7.64 \times 10^{-3}$** | $9.02 \times 10^{-2}$ | 0.674 |
| HG | STAI-Y-2 | 0.325 | 0.012 | $2.77 \times 10^{-2}$ | **$2.58 \times 10^{-3}$** | 0.285 |
| Hypoth. | STAI-Y-2 | 0.260 | 0.080 | **$2.89 \times 10^{-3}$** | 0.147 | 0.0979 |
| Thalamus | STAI-Y-2 | 0.167 | 0.198 | **$4.86 \times 10^{-3}$** | $9.24 \times 10^{-2}$ | 0.555 |
| Hippocamp. | STAI-Y-2 | 0.161 | 0.060 | **$5.64 \times 10^{-3}$** | 0.615 | 0.266 |
| IC | STAI-Y-2 | 0.216 | 0.154 | **$6.56 \times 10^{-3}$** | $1.46 \times 10^{-2}$ | 0.124 |
| AC | COMPASS | 0.066 | 0.024 | **$2.77 \times 10^{-3}$** | 0.237 | 0.902 |
| Thalamus | COMPASS | 0.107 | 0.024 | **$6.21 \times 10^{-3}$** | 0.249 | 0.296 |

most significant effect is that of the group (HC vs FM) in the hypothalamus, thalamus, hippocampus, and IC. The initial rise depends on STAI-Y-2 scores in the HG only, and no regions showed significant interaction effects. These observations indicate that the initial rise is not related to STAI-Y-2 scores, but there are differences between the groups that are better identified when variations in STAI-Y-2 are accounted for. Accounting for variations in COMPASS-31 scores there are main effects of the group in the AC and thalamus. There are no significant variations with COMPASS-31 scores and no significant interaction effects.

## Results from additional data sets

BOLD time-course responses were extracted from the same regions/sub-regions as shown in Fig 3, with data obtained in participants with PVD, and the corresponding healthy control group, and are shown in Fig 6. The data were obtained under the same conditions as for the FM and HC group data described above. The time-course responses are plotted with the same vertical scales and are shown to be highly consistent across the PVD and HC groups, with notable differences in the responses across regions. BOLD signal changes are apparent at the time that participants were informed of impending stimulus, and brief increases in signal are visible in most regions after the onset and offset of stimulation. The results in Fig 6 for the HC group are also consistent with those shown in Fig 3. Measures of pupil area variations during fMRI acquisitions during the Pain condition are shown in Fig 7. Values are shown averaged over groups of females with FM, and a corresponding healthy control group. The average pupil area is 1540 pixel units in the HC group, and 1920 pixel units in the FM group, demonstrating that, on average, the FM group had approximately 25% larger pupil area than the HC group.

## Discussion

### Connectivity analysis

SAPM results demonstrate relationships between input and output signaling (i.e. connectivity values) that are generally similar in FM and HC groups, except as noted below. There were more differences identified in the thalamus, hypothalamus, and brainstem regions than in

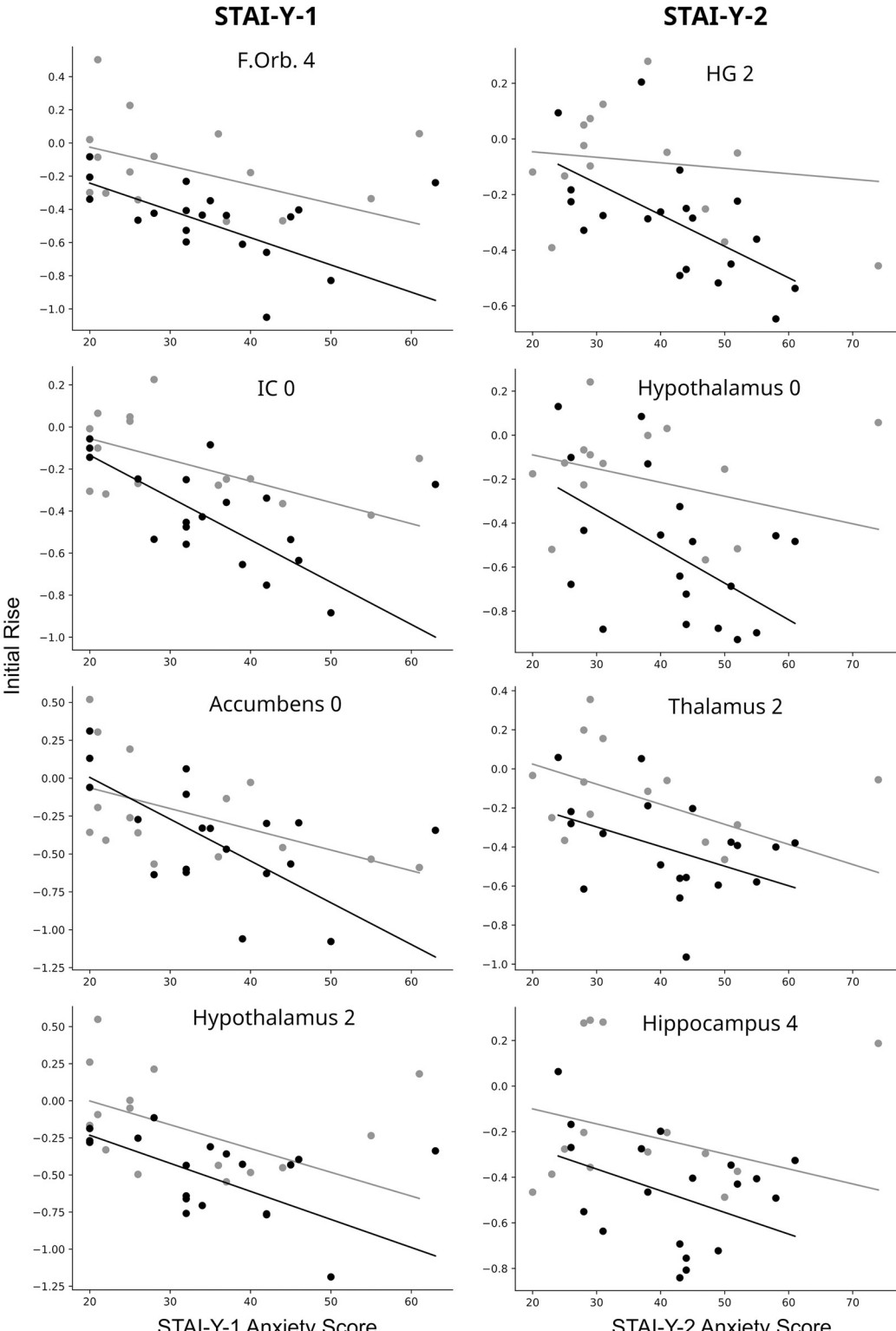

**Fig 5. Selected plots of the initial rise (time-course starting point) versus questionnaire scores including the state anxiety score (STAI-Y-1) in the left column, the trait anxiety score (STAI-Y-2) in the right column.** Black symbols indicate data for the FM group and gray symbols show the HC group data. Plots are included for regions which had significant (p < 0.01) values for either a main effect or interaction effect identified by means of a 2 x 2 ANCOVA, as listed in Table 3.

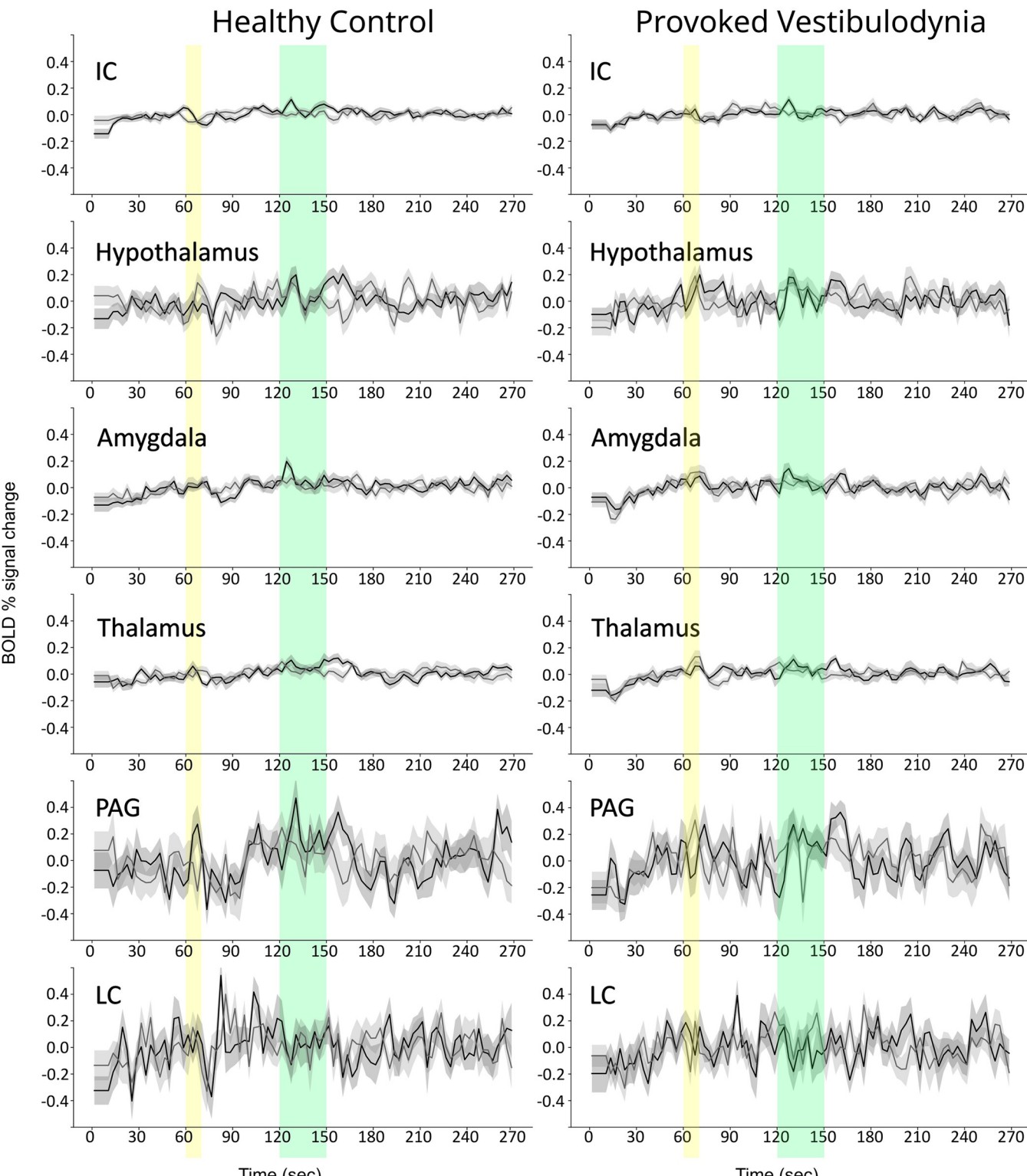

**Fig 6. Comparisons of BOLD time-course responses for comparison with the results in Fig 3.** Results from healthy participants (HC) are shown in the left column and results from participants with provoked vestibulodynia (PVD) and in the right column. Black lines indicate BOLD responses for runs with noxious stimulation, and gray lines show responses to runs without stimulation. The period when participants were informed of the stimulus type is indicated with a yellow band, and the stimulation period is indicated with a green band.

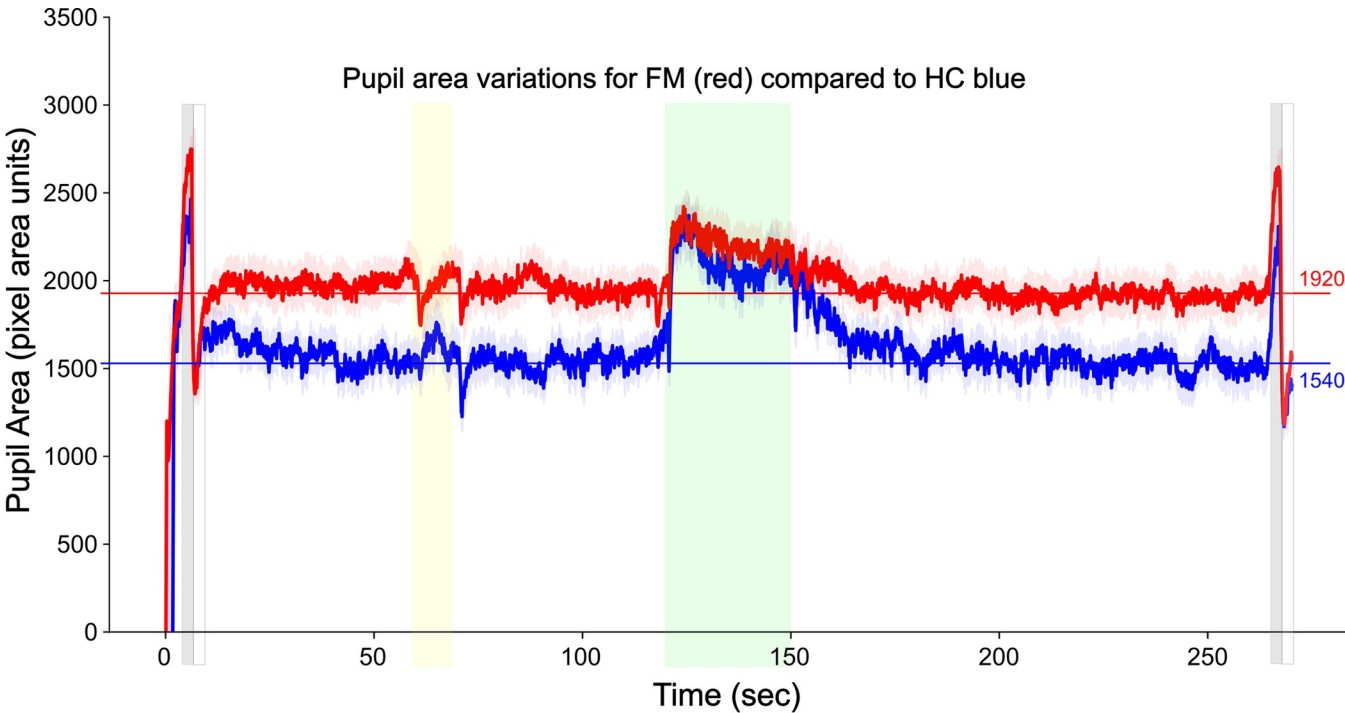

**Fig 7. Plots of pupil areas recorded during fMRI acquisitions during the pain condition.** Values are averaged over 125 trials in 26 participants with FM (red lines), and 45 trials in 10 healthy control participants (blue lines). At the start and end of each fMRI run participants viewed a completely white screen followed by a completely black screen to provide pupil size references. These periods are indicated with gray- and white-filled boxes. The period when participants were informed of the impending stimulus is indicated with the yellow band, and the stimulation period is indicated with the green band.

cortical regions. This is consistent with previous fMRI investigations of FM [13, 16]. In the present study we observed that some connections may reflect the constant pain that is experienced by the FM participants, including during the No-Pain condition. This is suggested by connectivity values in the FM group for both Pain and No-Pain study conditions that were more similar to those observed during the Pain condition in the HC group. These connections include the IC→AC, AC→FOrb, thalamus→hypothalamus, amygdala→hippocampus, LC→thalamus, and hypothalamus→thalamus connections. Another notable trend is that 12 of the 18 connections (Table 1) have lower magnitude connectivity values in FM compared to HC. However, the NRM→C6RD connection had consistently stronger connectivity values (in magnitude) in the FM group than the HC group, regardless of the study condition. These results indicate that although the connectivity values in the FM and HC groups are similar, there appear to be differences between the groups independent of the study condition. The results are consistent with our previous studies which indicated that there are altered motivational and affective responses in FM compared to HC (via cortical regions) and altered brain stem modulation of responses to nociceptive stimuli (via the NRM→C6RD pathway [13, 21, 24]. In FM participants the NRM→C6RD connection was consistently negative (Table 1), indicating that it provides inhibitory signaling to the cord dorsal horn, but this effect is not observed in HC. This altered descending regulation may be a response to central sensitization in the spinal cord as has been suggested previously [36, 58], and as evidenced by the higher heat pain sensitivity in FM.

SAPM results also demonstrate effects that differ between the Pain and No-Pain conditions, and may not be caused by chronic pain in the FM group. In both groups, the PBN→thalamus and hypothalamus→thalamus connections had consistently higher connectivity values in the

Pain condition compared to No-Pain, whereas the LC→PBN connection had lower connectivity values in Pain compared to No-Pain. These results show that the differences observed between FM and HC groups cannot be attributed solely to the chronic pain experienced by participants with FM. The ANCOVA results in Table 2 show that some connectivity values differed between groups in relation to personal characteristics such as pain ratings, autonomic symptoms, and anxiety and pain catastrophizing scores. These results may identify differences in neural signaling in FM, in terms of the apparent transmission effect (i.e. how each input signal is transformed to contribute to the output signaling from a region) [26].

## BOLD response characteristics

While the connectivity values demonstrate the structure of the brainstem/cord network and the influences of signaling between regions, it is the BOLD signal variations that demonstrate how this signaling varies over time and in response to external influences such as heat stimuli and emotional or cognitive influences. BOLD signal variations observed in each sub-region, combined with the differences in connectivity values, reveal important features of how neural signaling is altered in FM. The properties of BOLD responses shown in Fig 3 demonstrate the overall consistency of responses between the Pain and No-Pain conditions, with the notable exceptions at the times when participants were informed of the stimulus type, and when the stimulus was applied. In almost every region shown in Fig 3 there are clear responses after the onset and offset of stimulation. These are BOLD responses to the change in condition and are inferred to relate to a change in cognitive state, possibly related to novelty or salience. Comparisons between FM and HC groups during the Pain condition show similar increases in BOLD signal after the onset and offset of stimulation. However, another prominent feature of the BOLD response in each region is the initial rise in signal during approximately the first 40 seconds of each run. This rise is observed in every region shown in Fig 3 and is approximately twice the magnitude in FM compared to HC. The BOLD responses after the onset and offset of stimulation therefore reach a higher peak level in the FM group, relative to the signal level at the start of each run. Similar to the responses at the onset and offset of stimulation, this initial rise is believed to be related to a change in cognitive state at the beginning of each fMRI run because it is not concurrent with the application of the heat stimulus.

## Interpretation of the results

There are two possible interpretations of the initial rise in BOLD signal. One is that the FM group begins at a lower level of metabolic demand in every region in our analysis, and is aroused to a "normal" state of metabolism each time an fMRI run begins. The second possibility is that the two groups begin at similar levels of metabolic demand, and the FM group rises to a heightened state of metabolic demand which is then sustained for the remainder of each fMRI run. This second interpretation is supported by the measurements of pupil areas that were obtained during a separate fMRI study (Fig 7). These measurements demonstrated pupil dilation at the beginning of each fMRI run, and sustained pupil areas that were approximately 25% larger on average in the FM group throughout the remainder of each run. Both groups had increases in pupil area during stimulation periods, but the peak pupil areas during stimulation were similar for the two groups. The BOLD responses and pupil area measurements are thus both consistent with prior observations that patients with FM are in a persistent state of heightened autonomic signaling [59]. It could similarly demonstrate other, non-autonomic, signaling as well. The BOLD signal variations in the brain were observed to occur in a number of regions that are known to be involved with pain processing, but many of the regions shown in Fig 3 including the IC, hypothalamus, and PAG, are also components of an autonomic

signaling network as described by Craig [60]. The effects of altered autonomic signaling can therefore be expected to influence signaling across the entire pain network. The interconnected effects of autonomic signaling and emotional and cognitive influences cannot be distinguished in the results of the present study in terms of how they contributed to, or were affected by, the pain that resulted from the noxious heat stimuli. The observed BOLD responses and pupil measurements are both consistent with the conclusion that the results reflect heightened signaling across interconnected networks of regions in FM, compared to the HC group, during each fMRI run.

A third possible explanation for the initial rise that must be considered is that it is the result of errors or artifacts, and is not related to physiological effects. The fact that the initial rise does not occur in two sets of results from HC participants (from the FM and PVD studies) and does not occur in the PVD group provides evidence that the initial rise is not a systematic error or artifact arising from the data acquisition or analysis methods. MRI signal variations related to changes in relaxation-time weighting (T1-weighting) at the beginning of each run produce decreases in signal over time, not increases, and were avoided by masking the data from the first three volumes of each run. Moreover, differences in this effect between the FM and HC groups would require consistent differences in the tissue properties in the brainstem and spinal cord and correspondingly different T1 values. This would be an important finding, but there is no evidence of this difference in tissue properties occuring. Finally, effects of head motion can be ruled out as a cause because records of motion obtained from co-registering (i.e. motion correcting) each fMRI run do not show any evidence of consistent motion effects in either the FM or HC groups. We therefore conclude that the observed initial rise is physiological in nature and is a BOLD signal response.

The cause of the initial rise was investigated by comparing its size with each participant's pain sensitivity, questionnaire scores, and between participants taking SNRIs, or not. With data stratified according to pain sensitivity, there is no evidence of differences in the magnitude of the initial rise in relation to differences in pain sensitivity (S2 Appendix). The size of the initial rise was observed to be consistent across repeated fMRI runs within each imaging session, and does not appear to be affected by SNRIs. In data from women with PVD, the initial rise was similar to that observed in HC groups. The only factors that appear to be related to the size of the initial rise (summarized in Table 3 and Fig 5) are state and trait anxiety scores. ANCOVA results show that while the size of the initial rise varies in relation to STAI-Y-1 and STAI-Y-2 scores in some regions, differences in anxiety scores between FM and HC groups do not explain the difference in initial rise between the groups. This conclusion is based on the observation that for the same STAI-Y-1 or STAI-Y-2 scores the HC and FM participants do not have the same initial rise. However, in several regions including the IC, nucleus accumbens, and amygdala, the relationships between the size of the initial rise and STAI-Y-1 scores appear to converge for the HC and FM groups at lower anxiety scores.

These results are consistent with the conclusion that the size of the initial rise is related to each participant's anxiety. However, the effect is exaggerated in FM. The STAI-Y-1 scores relate to fear responses and arousal of the autonomic nervous system that are caused by stressful situations or perceived threats [27]. The initial rise is therefore likely related to stress or fear at the onset of each fMRI run. It is notable that the effect does not diminish across multiple repeated runs. Since the response occurs across many regions of the brain it may be caused by volume transmission such as norepinephrine, cortisol, or serotonin. Previous studies have identified an important role of altered signaling to/from the locus coeruleus in FM [13], suggesting that norepinephrine may play a role. However, no effect of SNRIs was detected in this study group to support this concluson.

The presence of the initial rise, and its consistent occurrence across repeated fMRI runs, requires that the effect vanishes between fMRI runs. It is therefore sustained for periods of minutes during each fMRI run, and takes only minutes to dissipate. Participants were familiarized with the study procedures prior to the MRI session, and were informed via visual and auditory cues when an imaging session was about to begin, and also by the sounds of the MRI system when imaging was in progress. While it is possible that the sounds of the MRI system during imaging evoked fear or stress and explain the rise in BOLD signal in the FM group, we note that the stimulus temperatures were calibrated during the training session and pain ratings were consistent between the training session (without any MRI sounds) and the fMRI session. Nonetheless, it is possible that each fMRI run created a mentally stressful environment for the participants. It is expected that a cognitive state related to factors such as fear or expectation would vary over repeated runs such as diminishing with familiarity. The timing of the effect and the fact that it is widespread across brain regions suggests that the initial rise does not reflect a cognitive state. It is consistent with the effect being mediated by a hormone, such as norepinephrine, cortisol, or serotonin. The observed BOLD signal characteristics may therefore reflect the hormonal dysfunction described previously, possibly as a result of hyperactivity of the HPA axis [2–6]. Demori et al. [10] have recently described a theoretical model of FM which is stress-related and involves an altered stable balance of the HPA axis and the hypothalamus-pituitary-gonadal (HPG) axis which alters the balance of glutamate and GABA in pain processing.

## Limitations

The SAPM method is inherently limited because it models complex systems of interconnected regions of the central nervous system as a simplified linear model in order to aid our understanding of neural signaling within such systems. The linear approach is necessary in the absence of evidence for how to model it as a more complex non-linear system, and represents a first-order approximation of signaling across the network. The advantage of the linear modeling approach is that it requires fewer fit parameters and is less likely to suffer from overfitting. Moreover, with the SAPM method we model the average effects of signaling between regions/sub-regions, again in order to simplify the model, avoid over-ftting, and facilitate our understanding of the results. Another form of simplifcation is that the model is not complete, as it involves only selected regions within the volume spanned by the data, and it does not include signaling within each region. Nonetheless, the method has been shown to provide valuable insights into neural signaling across networks and highly consistent results. The results have been validated previously in relation to known properties of neural signaling [25]. However, as with any new method, SAPM is still limited because it is not yet used by multiple researchers and requires further validation and testing, which is one of the purposes of the present study. The present study demonstrates that the SAPM approach provides a much more detailed picture of neural signaling and how it is altered in FM than can be achieved by other methods, in spite of the limitations.

## Summary and conclusions

Together, the analyses of BOLD timecourse responses and connectivity values provide a detailed description of neural signaling related to pain, and how it is altered in FM. For example, the BOLD responses in the IC may be the result of latent input, possibly via afferent autonomic input as described by Craig [60], and by excitatory input from the AC as indicated by the SAPM connectivity results. The AC→IC connectivity values are lower in FM than in HC participants meaning that input signaling from the ACC causes greater output signaling from

the IC in the HC participants. The initial rise in BOLD signal is larger in the IC in people with higher state anxiety scores (STAI-Y-1), and the effect is more exaggerated in people with FM. Similarly, the SAPM results show evidence of the hypothalamus receiving excitatory input from the thalamus, and its effect on the hypothalamus output is reduced in FM compared to HC, and also reduced in the HC participants during the noxious stimulation condition. The hypothalamus in turn provides input signaling to the PAG and NTS and may therefore influence descending pain regulation. The initial rise in BOLD signal is also larger in the hypothalamus in people with higher state anxiety scores, regardless of whether they have FM or not.

The key findings of this study show that the initial rise and persistently elevated BOLD signal and the concurrent pupil responses may be important indicators of the mechanisms underlying FM. These observations are consistent with previous studies showing that FM is related to stress, and altered neurohormonal regulation and/or autonomic regulation, likely involving hyperactivity of the HPA axis [2–6, 9]. People with FM appear to have exaggerated responses to any stressor, such as being subjected to mildly noxious stimulation, even after being familiarized with the procedures and experiencing repeated runs. This limits their ability to respond further to additional stressors. Network modeling with SAPM indicates that, in general, the relationships between input and output signaling are similar in FM and HC, but that neural signaling between regions is altered in FM.

The results may illustrate the fact that pain is an emotional and cognitive response to noxious stimuli and that the responses in cortical regions involve predominantly the higher-level processes of assessing the pain and its future implications [61]. This assessment depends on past experiences, the person's current situation and environment, autonomic feed-back from the body, and the intensity and modality of the noxious stimulus. Descending pain regulation is modified by these factors via the brainstem and by reciprocal feed-back between the spinal cord and brainstem. It therefore appears that in people with FM, interactions between anxiety, autonomic responses, and pain, produce a self-sustaining state of heightened sensitivity. This state may be demonstated by the observed hypervigilence in people with FM. Future studies are required to determine whether or not the findings of this study are specific to people with FM, or are a common feature of chronic pain conditions, and how these findings may be used to develop improved diagnostic methods and treatments for FM.

## Supporting information

**S1 Appendix. Comparisons of BOLD responses detected with data from the brain and from the brainstem/spinal cord.**
(DOCX)

**S2 Appendix. Details of all ANCOVA comparisons between study groups (FM vs HC), connectivity values and the magnitude of the initial rise in BOLD signal, in relation to questionnaire scores.**
(DOCX)

**S3 Appendix. Comparisons of BOLD responses for participants stratified according to pain sensitivity.**
(DOCX)

## Acknowledgments

We gratefully acknowledge the contributions of Gabriela Ioachim, Jocelyn Powers, Howie Warren, and Lindsey Yessick for conducting the original studies which provided the fMRI data, and Shima Hassanpour, Hannan Algitami, Maya Umraw, Jessica Merletti, and Brieana

Keast for contributing to the study which provided the eye-tracking data. None of the authors have any conflicts of interest to declare. The data used in this study are available upon request from the corresponding author, for ethical reasons. The software used for analysis is freely available on GitHub (www.github.com) as detailed in the Methods section.

## Author Contributions

**Conceptualization:** Patrick W. Stroman, Roland Staud, Caroline F. Pukall.

**Data curation:** Patrick W. Stroman, Caroline F. Pukall.

**Formal analysis:** Patrick W. Stroman.

**Funding acquisition:** Patrick W. Stroman.

**Investigation:** Patrick W. Stroman, Caroline F. Pukall.

**Methodology:** Patrick W. Stroman, Roland Staud.

**Project administration:** Patrick W. Stroman.

**Resources:** Patrick W. Stroman.

**Software:** Patrick W. Stroman.

**Supervision:** Patrick W. Stroman, Caroline F. Pukall.

**Validation:** Patrick W. Stroman.

**Visualization:** Patrick W. Stroman, Roland Staud, Caroline F. Pukall.

**Writing – original draft:** Patrick W. Stroman.

**Writing – review & editing:** Patrick W. Stroman, Roland Staud, Caroline F. Pukall.

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
