## [Decision Letter · Decision Letter 0]

30 Aug 2024

PONE-D-24-17827Evidence of a persistent altered neural state in people with fibromyalgia syndrome during functional MRI studies and its relationship with pain and anxietyPLOS ONE

Dear Dr. Stroman, Thank you for submitting your manuscript to PLOS ONE. After careful consideration, we feel that it has merit but does not fully meet PLOS ONE’s publication criteria as it currently stands. Therefore, we invite you to submit a revised version of the manuscript that addresses the points raised during the review process. Please submit your revised manuscript by Oct 14 2024 11:59PM. If you will need more time than this to complete your revisions, please reply to this message or contact the journal office at plosone@plos.org. Please include the following items when submitting your revised manuscript:A rebuttal letter that responds to each point raised by the academic editor and reviewer(s). You should upload this letter as a separate file labeled 'Response to Reviewers'.A marked-up copy of your manuscript that highlights changes made to the original version. You should upload this as a separate file labeled 'Revised Manuscript with Track Changes'.An unmarked version of your revised paper without tracked changes. You should upload this as a separate file labeled 'Manuscript'.

We look forward to receiving your revised manuscript.

Kind regards,

Phakkharawat Sittiprapaporn, Ph.D.

Academic Editor

PLOS ONE

Journal Requirements:

"Canadian Institutes of Health Research, Project Grant, 2023-2028: PWS, RS, CFP Natural Sciences and Engineering Research Council of Canada, RGPIN-2020-06777: PWS"

4. Please note that funding information should not appear in the Acknowledgments section or other areas of your manuscript. We will only publish funding information present in the Funding Statement section of the online submission form. Please remove any funding-related text from the manuscript. 

5. We note that you have indicated that there are restrictions to data sharing for this study. For studies involving human research participant data or other sensitive data, we encourage authors to share de-identified or anonymized data. However, when data cannot be publicly shared for ethical reasons, we allow authors to make their data sets available upon request. For information on unacceptable data access restrictions, please see http://journals.plos.org/plosone/s/data-availability#loc-unacceptable-data-access-restrictions. 

6. In this instance it seems there may be acceptable restrictions in place that prevent the public sharing of your minimal data. However, in line with our goal of ensuring long-term data availability to all interested researchers, PLOS’ Data Policy states that authors cannot be the sole named individuals responsible for ensuring data access (http://journals.plos.org/plosone/s/data-availability#loc-acceptable-data-sharing-methods).

**Additional Editor Comments:**

Your manuscript, referenced above, has now been reviewed by experts in the field. After careful consideration, we feel that it has merit but does not fully meet PLOS ONE’s publication criteria as it currently stands. Specifically, experiments, statistics, and other analyses must be performed to a high technical standard and described in sufficient detail. The reviewers have made some suggestions, which the Editor feels would improve your manuscript. We encourage you to consider these comments and make an appropriate revision of your manuscript. Therefore, we invite you to submit a revised version of the manuscript that addresses the points raised during the review process. The comments of the reviewers are included below in order for you to understand the basis for our decision, and we hope that their thoughtful comments will help you in your revision.

Reviewers' comments:

Reviewer's Responses to Questions

**Comments to the Author**

1. Is the manuscript technically sound, and do the data support the conclusions?

Reviewer #1: Yes

Reviewer #2: Yes

2. Has the statistical analysis been performed appropriately and rigorously? 

Reviewer #1: I Don't Know

Reviewer #2: Yes

3. Have the authors made all data underlying the findings in their manuscript fully available?

Reviewer #1: No

Reviewer #2: Yes

4. Is the manuscript presented in an intelligible fashion and written in standard English?

Reviewer #1: Yes

Reviewer #2: Yes

5. Review Comments to the Author

Reviewer #1: Stroman et al. examined task-fMRI data from a painful stimulation paradigm in women with fibromyalgia and healthy females using a novel approach developed by the authors. The way I understood it, this approach tests connectivity weights of input and output signals in an anatomically informed network model. They found differences between groups and associations with questionnaires. The main finding was a signal increase at the beginning of each scan that was more pronounced in women with fibromyalgia. This signal increase was associated with anxiety. In a similar sample the authors also found larger pupil sizes in women with fibromyalgia. They conclude that people with fibromyalgia may have an exaggerated stress response.

More research about the pathophysiology of fibromyalgia is needed. The present study is interesting, and the reasons to not make the underlying data publicly available are sound. However, I have several comments and questions:

Although previous work is referenced for characterization of the study sample, a summary of key demographic data should be presented in this manuscript as well.

The authors report that scans took place in two different (albeit very similar) scanners or scanner versions. Were the different scanners accounted for in the analyses? What proportion of the data was acquired in each configuration?

For normalization, briefly describe the normalization approach. Reference the original articles for the applied approach, not your own review discussing different approaches.

Which atlas(es) were used to select the anatomical regions?

It is not clear to me whether just one subregion per region was used in all analyses. The subregions that were used for SAPM should be identified for each region (e.g. in a supplement).

The authors explain what D-values are in SAPM, but an explanation for B-values is missing.

Which software (and its version) was used to perform statistics after SAPM, e.g. for correlations and ANCOVAs?

For ANCOVAs the models and the results for all tested factors and interactions should be reported at least in a supplement. Additionally the models should be presented in methods. This information is necessary to assess the rigor of the statistical analyses.

In general, no “omissions for brevity” should be made without being presented in a supplement.

The authors should add a discussion of the limitations of their study.

A brief discussion whether the initial rise could be a technical artifact or not might back up the conclusions.

Reviewer #2: In the manuscript “Evidence of a persistent altered neural state in people with fibromyalgia syndrome…” the authors reanalyze several prior sets of data using a novel analytic technique (SAPM). The main finding is that the main change in responsiveness noted appears related to the fMRI environment in FM more than the warning about noxious stimulation or the stimulation itself. The authors postulate, with correlation data with anxiety questionnaire measures, that this consistent with a sympathetic autonomic response to anxiety or stress throughout the fMRI acquisitions.

The authors suggest that the data presented in Figure 2 justify combining disparate methods together for this analysis. To my eyes, I am not sure that assumption is valid. Using some sort of metric to demonstrate intercompatibility would be desirable here.

The connectivity analysis, confined to network figures and tables, seems very fair. Statistics seem appropriate and the data seems useful. The rest of the manuscript is heavy with noisy waveform figures, the purpose of which is not always easy to intuit. It is also not an easy read.

The finding of most interest to the authors is displayed in Figure 5. The rapid increase in BOLD activity across many regions of the network discussed in Figure 1 occurs just from starting the task, independent from pain cues or pain stimulus. This is interesting and it relates to anxiety scores. It is also much greater than the changes that come with pain cues or pain stimulus, which is interesting consider recent ideas of central sensitization focus on overactive nociceptive responsiveness. The authors then observe that this pre-task BOLD increase is not a feature of a focal pain disorder (PVD). The authors then report pupil dilation data in a single line. Again, the readers are asked to eyeball the group differences without any metrics to help calculate group differences. The authors discuss that they interpret this eye tracking data to be persistent autonomic tone (perhaps constantly driving the oculomotor nerve to stay dilated); however persistent autonomic tone is not what they observed in the brain. This divergence seems to merit some further elaboration.

The authors note that this effect returns to a baseline level between fMRI runs. They do not clarify what that means in regards to the participant. Assuming that these participants remained in the magnet between runs, what was the stimulus that noted to the participant that a new run was starting. Was it a verbal cue? Was it the sound of the magnet starting up? If there is no discernable cue between runs, the finding takes on a different meaning. This warrants elaboration.

The manuscript feels like two separate manuscripts at times. The BOLD results are not placed in the context of the network connections that the authors’ painstakingly drew. Does knowing the network pathways elicited earlier in the manuscript predict where the BOLD phenomenon occurs as described later in the manuscript? Conversely, what does the network of BOLD activation the authors observed look like? How do these two different types of data inform each other?

One of the issues with re-analysis of data is that it is often hard for the audience to understand how the presented findings are different from the work that came before. Without requiring the audience to review each citation, it is not clear what the results of prior analysis of this data revealed. This is worth a paragraph in the discussion.

The authors conclude ‘The findings may relate to the well-known global hypersensitivity of FM participants’. The authors may consider going further with what their data suggests about both fibromyalgia and central sensitization. How do these data fit in with the popular idea of central sensitization that is implicated in “osteoarthritis, rheumatoid arthritis, chronic fatigue syndrome, Ehlers-Danlos syndrome, tendonopathies, headaches, spinal pain…” (Nijs, Lancet Rheumatology 2021)? Perhaps not very well.

The introduction does set the stage for this, noting all the studies in which the brain’s pain responsiveness is not particularly different between FM and HVs. These results suggest that fibromyalgia is more akin to an anxiety disorder than a pain disorder. The largest functional brain changes that occur during a pain task are unrelated to actually having pain. Persons with a more painful phenotype had less brain activation to painful stimuli compared to those with a less painful phenotype. These data do not support the idea of fibromyalgia as a disorder of hyperactivity to painful stimulation or sensitization of the pain network to incoming stimuli. It seems that being more direct about the implications of the findings would be worth considering in the discussion.

6. PLOS authors have the option to publish the peer review history of their article (what does this mean?). If published, this will include your full peer review and any attached files.

Reviewer #1: No

Reviewer #2: No

---

## [Author Response · Author response to Decision Letter 0]

11 Oct 2024

The responses to reviewers are also included in an uploaded document.

Responses to Reviews from PLOS One (PONE-D-24-17827R1)

We appreciate the time taken by the reviewers. We found the comments and suggestions to be very helpful and we believe that the revisions have improved the quality and clarity of the manuscript. We have attempted to fully address each of the reviewers’ points, as detailed below. However, most of the points raised suggested that we add information or details, resulting in a substantial increase in the length of the manuscript. We have attempted to provide the suggested information but in places we felt that we had to balance the responses with keeping the manuscript focused on the topic and as concise as possible. We have also moved some information to supplementary material as suggested, and to reduce the length of the primary text.

Please find the detailed responses to each point below, following the reviewers’ comments (in italics) which we have included to be clear which points were are responding to for each revision.

Please note that the data used in this study are now freely available on FigShare in two repositories because of size limitations. The data can be found in “Brain fMRI data comparing pain responses to noxious heat in healthy controls (HC) for comparison with fibromyalgia syndrome (FMS)” at https://doi.org/10.6084/m9.figshare.27105808.v1 , and “Brain fMRI data comparing pain responses to noxious heat in fibromyalgia syndrome (FMS)” at https://doi.org/10.6084/m9.figshare.27103864.v1 .

Reviewer #1: Stroman et al. examined task-fMRI data from a painful stimulation paradigm in women with fibromyalgia and healthy females using a novel approach developed by the authors. The way I understood it, this approach tests connectivity weights of input and output signals in an anatomically informed network model. They found differences between groups and associations with questionnaires. The main finding was a signal increase at the beginning of each scan that was more pronounced in women with fibromyalgia. This signal increase was associated with anxiety. In a similar sample the authors also found larger pupil sizes in women with fibromyalgia. They conclude that people with fibromyalgia may have an exaggerated stress response.

More research about the pathophysiology of fibromyalgia is needed. The present study is interesting, and the reasons to not make the underlying data publicly available are sound. However, I have several comments and questions:

Although previous work is referenced for characterization of the study sample, a summary of key demographic data should be presented in this manuscript as well.

In the “Participants” subsection of the Methods we have included information about the participants being all female, and the range and average ages for the HC and FM groups. We have now added more information about the pain ratings and temperatures of the stimuli as well, as follows:

“As described previously (1), the calibrated temperature was significantly different (p = 2.55 x 10-5) between the two groups and averaged 50.89 ± 1.04 °C for the HC group and 47.51 ± 2.72 °C for the FM group. The average pain ratings to the last contact in each set of 10 contacts were 38.93 ± 12.07 and 44.22 ± 12.04 on a 0-100 scale, for the HC and FM groups respectively, and were not significantly different. The temperatures and pain ratings demonstrate the higher heat pain sensitivity in the FM group.”

The authors report that scans took place in two different (albeit very similar) scanners or scanner versions. Were the different scanners accounted for in the analyses? What proportion of the data was acquired in each configuration?

We have added the following sentences to the Methods section to provide this information:

“Approximately midway through data collection, the MRI system was upgraded from a Siemens Trio to a Siemens Prisma. Test scans involving our experimental paradigm were performed on two participants before and after the upgrade, and the data quality were compared. No significant differences in BOLD activity between the datasets were found.”

For normalization, briefly describe the normalization approach. Reference the original articles for the applied approach, not your own review discussing different approaches. Which atlas(es) were used to select the anatomical regions?

We did not reference a review article. We referenced our article that provided a detailed description and validation of our methods (Stroman PW, Powers JM, Ioachim G. Proof-of-concept of a novel structural equation modelling approach for the analysis of functional magnetic resonance imaging data applied to investigate individual differences in human pain responses. Hum Brain Mapp. 2023;44(6):2523-585 42.). It is necessary to rely on cited work that provide essential background in order to keep a paper focused on the current topic. Nonetheless, in order to respond to the reviewer’s comment, we have added the following information to the Methods section:

“The reference images were created by combining the MNI152 template from the Statistical Parametric Mapping (SPM12) software package (38) and the PAM50 template, as described by De Leener et al. (39). Anatomical region-of-interest maps were defined from multiple sources, including anatomical descriptions, freely shared anatomical maps, and the CONN15e software package (40-50). These sources were combined to create a single anatomical map. The normalization procedure employed different methods for the brain, and for brainstem and cord regions. For brain regions, the normalization process uses the python package “dipy” (https://dipy.org/documentation/1.5.0/documentation/ ) which is based on the ANTs (Advanced Normalization Tools) software (51, 52). Normalization of brainstem and spinal cord regions has been described previously (26, 53) and involves mapping sections of the template to the image data for regions with distinct anatomical features. These region positions then guide the identification of successive sections of the spinal cord based on cross-correlation, progressing away from the brainstem, in a manner that maintains the distance along the cord. The normalization procedure is based on the premise that the cord anatomy is likely much more consistent in size across people, than is the spine anatomy (44). The normalization was then fine-tuned using the Medical Image Registration (MIRT) toolbox, translated into python (54).”

It is not clear to me whether just one subregion per region was used in all analyses. The subregions that were used for SAPM should be identified for each region (e.g. in a supplement).

We state in the Methods section (lines 321 to 322) that:

“In order to compare results from the brain, and from brainstem/cord, the same sub-regions were used for SAPM analyses, for the regions that are common to both network models.”

To be more clear we have expanded this to read:

“This combination of sub-regions therefore has BOLD time-series responses that fit the network model better than other combinations of sub-regions. This same set of sub-regions was then used for all subsequent analyses for consistency. In order to compare results from the brain, and from brainstem/cord, the same sub-regions were used for SAPM analyses for the regions that are common to both network models.”

Showing the 3D anatomical subregions for all 14 regions that were included in the brain network, and the additional 6 regions in the brainstem/cord network that were not included in the brain would mean that we need to display 20 regions. Multiple images are needed to depict the spatial extent of 3D regions or it is necessary to generate multiple 3D views. We have attempted to show images of the anatomical regions previously (2, 3). We do not feel that this information is necessary to present our current results and that the time and effort it would take to generate the images is not justified.

The authors explain what D-values are in SAPM, but an explanation for B-values is missing.

We have corrected the sentence spanning lines 305-306 to explain the B values and it now reads:

“The B values reflect how the incoming signal is converted to contribute to the output signal within each region. ”

Which software (and its version) was used to perform statistics after SAPM, e.g. for correlations and ANCOVAs?

The ANCOVA and correlation methods are based on mathematical formulas that do not differ between software or its versions. All statistical analyses were done in python. Correlations were computed using the “corrcoef” function in the “numpy” (version 1.23.5) module, and ANCOVAs were computed using the “anova_lm” function in the “statsmodels” (version 0.14.0) module. We do not feel that these details are useful information for the reader. Nonetheless, we have clarified how the statistical values were computed by adding the following sentence in the Methods:

“All statistical analyses were done in python using custom-written software using freely shared python modules including numpy and statsmodels.”

For ANCOVAs the models and the results for all tested factors and interactions should be reported at least in a supplement. Additionally the models should be presented in methods. This information is necessary to assess the rigor of the statistical analyses.

We have added the following to provide more information about the ANCOVA model that was used:

“Analyses of covariance (two-way ANCOVAs) were used to investigate how connectivity strengths (specifically DB values) varied in relation to the study groups (FM vs HC) and questionnaire scores from each participant”

We tested 52 different network connections for each questionnaire score. Reporting the results for all combinations of all tested connections and all questionnaire scores is not practical. We feel that reporting the significant results provides a sufficient level of detail to judge the rigor of the analysis, and to effectively communicate the key results to the reader.

In general, no “omissions for brevity” should be made without being presented in a supplement.

There are only two places in the manuscript where we state that we show selected information “for brevity”. One is when reporting the significant connections identified with ANCOVA analyses (Table 3) and the other is when we report which factors appeared to be related to the magnitude of the initial rise on BOLD signal. In both of these instances we are comparing fMRI results (connectivity values, BOLD changes) to the behavioral data obtained from each person in the form of pain ratings and questionnaire scores. We do not see the value of reporting the results which show no relationships when they would obscure the results that appear to show significant relationships. This addition would only add significant length to the paper and supplementary material without adding to its clarity. Reducing the readability would only reduce the impact of this work.

The authors should add a discussion of the limitations of their study.

We have added a limitations section to the Discussion as follows:

“Limitations

The SAPM method is inherently limited because it models complex systems of interconnected regions of the central nervous system as a simplified linear model in order to aid our understanding of neural signaling within such systems. The linear approach is necessary in the absence of evidence for how to better model more complex non-linear systems, and represents a first-order approximation of signaling across the network. The advantage of the linear modeling approach is that it requires fewer fit parameters and is less likely to suffer from over-fitting. Moreover, with the SAPM method we model the average effects of signaling between regions/sub-regions, again in order to simplify the model, avoid over-fitting, and facilitate our understanding of the results. Another form of simplification is that the model is not complete, as it involves only selected regions within the volume spanned by the data, and it does not include signaling within each region. Nonetheless, the method has been shown to provide valuable insights into neural signaling across networks and highly consistent results. The results have been validated previously in relation to known properties of neural signaling. The present study shows that the SAPM approach provides a much more detailed picture of neural signaling and how it is altered in FM than can be achieved by other methods, in spite of the limitations.”

A brief discussion whether the initial rise could be a technical artifact or not might back up the conclusions.

We have add the following to the Discussion to address this point:

“A third possible explanation for the initial rise that must be considered is that it is the result of errors or artifacts, and is not related to physiological effects. The fact that the initial rise does not occur in two sets of results from HC participants (from the FM and PVD studies) and does not occur in the PVD group provides evidence that the initial rise is not a systematic error or artifact arising from the data acquisition or analysis methods. MRI signal variations related to changes in relaxation-time weighting (T1-weighting) at the beginning of each run produce decreases in signal over time, not increases, and were avoided by masking the data from the first three volumes of each run. Moreover, differences in this effect between the FM and HC groups would require consistent differences in the tissue properties in the brainstem and spinal cord and correspondingly different T1 values. This would be an important finding, but there is no evidence of this difference in tissue properties occuring. Finally, effects of motion artifacts, specifically in the FM group, can be ruled out as a cause because records of motion obtained from the co-registration step during the preprocessing of each fMRI run do not show any evidence of consistent motion effects in either the FM or HC groups. We therefore conclude that the observed initial rise is physiological in nature and is a BOLD signal response.”

Reviewer #2: In the manuscript “Evidence of a persistent altered neural state in people with fibromyalgia syndrome…” the authors reanalyze several prior sets of data using a novel analytic technique (SAPM). The main finding is that the main change in responsiveness noted appears related to the fMRI environment in FM more than the warning about noxious stimulation or the stimulation itself. The authors postulate, with correlation data with anxiety questionnaire measures, that this consistent with a sympathetic autonomic response to anxiety or stress throughout the fMRI acquisitions.

The authors suggest that the data presented in Figure 2 justify combining disparate methods together for this analysis. To my eyes, I am not sure that assumption is valid. Using some sort of metric to demonstrate intercompatibility would be desirable here.

In order to provide more information and clarify this point we have added the following text to the description, and we have moved this information into the supplementary material:

“For example, data sets obtained with the two methods demonstrate an initial rise in signal, as well as signal variations when participants were informed of what to expect and while anticipating the stimulus. This was followed by relatively low signal variations during the stimulation period. Direct comparison of the time-series responses were obtained by first interpolating the values to the sampling rate of the brainstem/cord data (TR = 6.75 sec, 40 time points) and allowing time shifts up to ½ of a TR period to account for differences in slice timing. The time-series responses with the two methods were observed have correlation, R, values of 0.263 and 0.446 for HC and FM data, respectively (corresponding to p = 0.051 and p = 0.0018). The data therefore demonstrate similar BOLD response features and the two methods are expected to provide similar results with SAPM.”

We have also added the following sentence to the caption for supplementary Figure S1 (formerly Figure 2):

“With data interpolated to the same TR interval as for the brainstem/cord data and allowing time 

---

## [Decision Letter · Decision Letter 1]

19 Nov 2024

PONE-D-24-17827R1Evidence of a persistent altered neural state in people with fibromyalgia syndrome during functional MRI studies and its relationship with pain and anxietyPLOS ONE

Dear Dr. Stroman,

Thank you for submitting your manuscript to PLOS ONE. After careful consideration, we feel that it has merit but does not fully meet PLOS ONE’s publication criteria as it currently stands. Therefore, we invite you to submit a revised version of the manuscript that addresses the points raised during the review process. Please submit your revised manuscript by Jan 03 2025 11:59PM. If you will need more time than this to complete your revisions, please reply to this message or contact the journal office at plosone@plos.org. Please include the following items when submitting your revised manuscript:A rebuttal letter that responds to each point raised by the academic editor and reviewer(s). You should upload this letter as a separate file labeled 'Response to Reviewers'.A marked-up copy of your manuscript that highlights changes made to the original version. You should upload this as a separate file labeled 'Revised Manuscript with Track Changes'.An unmarked version of your revised paper without tracked changes. You should upload this as a separate file labeled 'Manuscript'.If applicable, we recommend that you deposit your laboratory protocols in protocols.io to enhance the reproducibility of your results. Protocols.io assigns your protocol its own identifier (DOI) so that it can be cited independently in the future. For instructions see: https://journals.plos.org/plosone/s/submission-guidelines#loc-laboratory-protocols. Additionally, PLOS ONE offers an option for publishing peer-reviewed Lab Protocol articles, which describe protocols hosted on protocols.io. Read more information on sharing protocols at https://plos.org/protocols?utm_medium=editorial-email&utm_source=authorletters&utm_campaign=protocols.

We look forward to receiving your revised manuscript.

Kind regards,

Phakkharawat Sittiprapaporn, Ph.D.

Academic Editor

PLOS ONE

Journal Requirements:

Reviewers' comments:

Reviewer's Responses to Questions

**Comments to the Author**

1. If the authors have adequately addressed your comments raised in a previous round of review and you feel that this manuscript is now acceptable for publication, you may indicate that here to bypass the “Comments to the Author” section, enter your conflict of interest statement in the “Confidential to Editor” section, and submit your "Accept" recommendation.

Reviewer #1: (No Response)

Reviewer #3: All comments have been addressed

2. Is the manuscript technically sound, and do the data support the conclusions?

Reviewer #1: Yes

Reviewer #3: Partly

3. Has the statistical analysis been performed appropriately and rigorously? 

Reviewer #1: Yes

Reviewer #3: Yes

4. Have the authors made all data underlying the findings in their manuscript fully available?

Reviewer #1: Yes

Reviewer #3: Yes

5. Is the manuscript presented in an intelligible fashion and written in standard English?

Reviewer #1: Yes

Reviewer #3: Yes

6. Review Comments to the Author

Reviewer #1: The Authors have addressed most of my comments appropriately. However, I still disagree on a few points:

While the math behind correlations and ANCOVAs does not change, standard parameters (e.g. types of sums of squares, inclusion of an intercept) do in fact differ between software and sometimes versions. For the sake of reproducibility, functions and package versions should be provided (as is stated in the journal and linked guidelines).

I disagree on the notion that “ Reporting the results for all combinations of all tested connections and all questionnaire scores is not practical.” or that there would be no value in reporting the results not showing any relationships. If the analyses are worth doing, the results are worth to be reported. I agree that they should not be reported in the main text, hence the suggestion to put them into a supplement.

Minor:

Please provide the number of patients and the number of controls scanned in the trio configuration.

Please provide the number of participants for each group in the abstract. N=x in a bracket would be sufficient.

Did any of the patients take opioids or benzodiazepines? If so, please add the numbers to the participants section.

Reviewer #3: The authors responded to almost all the requests of the previous Reviewers. However, in my opinion, a clear reference to a possible practical application for patients with fibromyalgia syndrome is still missing in the conclusions of the updated manuscript. In light of the data presented, do the authors think that these results could be helpful for the future treatment of these patients? In which way? Drugs? Neurostimulation? New emerging techniques? Others?

I would also include among the limitations the analysis methodology used since no papers in the english literature (if we do not consider those of the authors themselves) have used this SAPM method for the analysis of fMRI datasets.

7. PLOS authors have the option to publish the peer review history of their article (what does this mean?). If published, this will include your full peer review and any attached files.

Reviewer #1: No

Reviewer #3: **Yes: **Cesare Gagliardo, University of Palermo

---

## [Author Response · Author response to Decision Letter 1]

21 Nov 2024

We have addressed all of the additional comments from the reviewers.

Please find the detailed responses to each point below, following the reviewers’ comments which we have included for clarity.

Reviewer #1: 

The Authors have addressed most of my comments appropriately. However, I still disagree on a few points:

While the math behind correlations and ANCOVAs does not change, standard parameters (e.g. types of sums of squares, inclusion of an intercept) do in fact differ between software and sometimes versions. For the sake of reproducibility, functions and package versions should be provided (as is stated in the journal and linked guidelines).

We have now added the following sentence to provide this information:

“Correlations were computed using the “corrcoef” function in the “numpy” (version 1.23.5) module, and ANCOVAs were computed using the “anova_lm” function in the “statsmodels” (version 0.14.0) module.”

I disagree on the notion that “ Reporting the results for all combinations of all tested connections and all questionnaire scores is not practical.” or that there would be no value in reporting the results not showing any relationships. If the analyses are worth doing, the results are worth to be reported. I agree that they should not be reported in the main text, hence the suggestion to put them into a supplement.

We have added supplementary tables (now S2 Appendix) showing detailed statistical values for all of the ANCOVA comparisons that were done.

Minor:

Please provide the number of patients and the number of controls scanned in the trio configuration.

We have now added this information as follows:

“Of the 20 FM participants included in this study, 15 were studied before the MRI upgrade, and 5 were studied after the upgrade. Of the 17 HC participants, 12 were studied before the upgrade, and 5 after.”

Please provide the number of participants for each group in the abstract. N=x in a bracket would be sufficient.

This information has been added to the abstract.

Did any of the patients take opioids or benzodiazepines? If so, please add the numbers to the participants section.

We already mention in the “Participants” section of the Methods that the participant characteristics, including medications, were discussed in detail in the original paper reporting this data set. We have now added additional information about medications, as follows:

“”Two of the 20 FM participants were taking opioid antagonists, one was taking opioid agonists, two were taking anxiolytics, and 17 of the 20 were taking antidepressants.”

Reviewer #3: 

The authors responded to almost all the requests of the previous Reviewers. However, in my opinion, a clear reference to a possible practical application for patients with fibromyalgia syndrome is still missing in the conclusions of the updated manuscript. In light of the data presented, do the authors think that these results could be helpful for the future treatment of these patients? In which way? Drugs? Neurostimulation? New emerging techniques? Others?

We do not believe that the current results provide enough information for us to comment on possible treatments or interventions. Nonetheless, the features of FM that we have identified are an important advance in our understanding of FM. We have attempted to address the reviewer’s question by adding the following sentence to the Conclusions:

“Future studies are required to determine whether or not the findings of this study are specific to people with FM, or are a common feature of chronic pain conditions, and how these findings may be used to develop improved diagnostic methods and treatments for FM.”

I would also include among the limitations the analysis methodology used since no papers in the english literature (if we do not consider those of the authors themselves) have used this SAPM method for the analysis of fMRI datasets.

We have added the following sentence to the limitations section:

“However, as with any new method, SAPM is still limited because it is not yet used by multiple researchers and requires further validation and testing, which is one of the purposes of the present study.”

---

## [Editor Report · Decision Letter 2]

16 Dec 2024

Evidence of a persistent altered neural state in people with fibromyalgia syndrome during functional MRI studies and its relationship with pain and anxiety

PONE-D-24-17827R2

Dear Dr. Stroman,

We’re pleased to inform you that your manuscript has been judged scientifically suitable for publication and will be formally accepted for publication once it meets all outstanding technical requirements.

Kind regards,

Phakkharawat Sittiprapaporn, Ph.D.

Academic Editor

PLOS ONE

---

## [Editor Report · Acceptance letter]

10 Jan 2025

PONE-D-24-17827R2 

PLOS ONE

Dear Dr. Stroman, 

I'm pleased to inform you that your manuscript has been deemed suitable for publication in PLOS ONE. Congratulations! Your manuscript is now being handed over to our production team.

Kind regards, 

on behalf of

Dr. Phakkharawat Sittiprapaporn 

Academic Editor

PLOS ONE